# Towards a coupled model to investigate wave-sea ice interactions in the Arctic marginal ice zone

Guillaume Boutin[1], Camille Lique[1], Fabrice Ardhuin[1], Clément Rousset[2], Claude Talandier[1], Mickael Accensi[1], and Fanny Girard-Ardhuin[1]

[1]Univ. Brest, CNRS, IRD, Ifremer, Laboratoire d'Océanographie Physique et Spatiale, IUEM, Brest, France
[2]Sorbonne Universités (UPMC Paris 6), LOCEAN-IPSL, CNRS/IRD/MNHN, Paris, France

**Correspondence:** boutinguillaume87@gmail.com

**Abstract.** The Arctic Marginal Ice Zone (MIZ), where strong interactions between sea ice, ocean and atmosphere are taking place, is expanding as the result of on-going sea ice retreat. Yet, state-of-the-art models exhibit significant biases in their representation of the complex ocean-sea ice interactions taking place in the MIZ. Here, we present the development of a new coupled sea ice-ocean wave model. This set up allows us to investigate some of the key processes at play in the MIZ. In particular, our coupling enables to account for the wave radiative stress resulting from the wave attenuation by sea ice, and the sea ice lateral melt resulting from the wave-induced sea ice fragmentation. We find that, locally in the MIZ, the ocean surface waves can affect the sea ice drift and melt, resulting in significant changes in sea ice concentration and thickness as well as sea surface temperature and salinity. Our results highlight the need to include wave-sea ice processes in models used to forecast sea ice conditions on short time scales. Our results also suggest that the coupling between waves and sea ice would ultimately need to be investigated in a more complex system, allowing for interactions with the ocean and the atmosphere.

## 1 Introduction

Numerical models exhibit large biases in their representation of Arctic sea ice concentration and thickness, regardless of their complexity or resolution (Stroeve et al., 2014; Chevallier et al., 2017; Wang et al., 2016; Lique et al., 2016). Comparing 10 reanalyses based on state-of-the-art ocean-sea ice models against observations, Uotila et al. (2018) found that model biases were the largest in the Marginal Ice Zone (MIZ). Indeed, the MIZ is characterized by a wide variety of processes resulting from the highly non-linear interactions between the atmosphere, ocean and sea ice: sea ice floe fragmentation and welding, lead opening and associated heat transfers, mesoscale and submesoscale features arising from strong temperature and salinity gradients (see Lee et al., 2012, for a review and references therein), and many of these processes are only crudely (if at all) taken into account in models. Some of these processes, sea ice fragmentation in particular, result from interactions between ocean surface waves and sea ice, and are thought to be key for the dynamics and evolution of the MIZ (Thomson et al., 2018). These interactions are the focus of the current paper.

Sea ice in the Arctic has been drastically receding over the past few decades (Comiso et al., 2017), resulting in an expansion of the MIZ in summer (Strong and Rigor, 2013) which is expected to intensify in the future (Aksenov et al., 2017). This provides an expanding fetch for waves to grow and propagate (Thomson and Rogers, 2014; Waseda et al., 2018), suggesting an overall increase of wave heights in the Arctic (Stopa et al., 2016). Once generated, waves can then propagate into sea ice, strongly impacting both dynamical and thermodynamical sea ice properties in the MIZ through different mechanisms (Asplin et al., 2012). First, observations suggest that waves determine the shape and size of the sea ice floes in the MIZ, through the fragmentation occurring when the ice cover is deformed (Langhorne et al., 1998), or by controlling the formation of frazil/pancake ice (Shen and Ackley, 1991). Wave-induced sea ice fragmentation is also expected to affect lateral melt (Steele et al., 1992), heat fluxes between ocean and atmosphere (Marcq and Weiss, 2012), and sea ice drift in the MIZ (McPhee, 1980; Feltham, 2005; Williams et al., 2017). When breaking in the MIZ, waves can generate turbulence in the mixed layer (Sutherland and Melville, 2013), possibly affecting the rate of ice formation or melting by modulating heat fluxes between the ocean, the sea ice and the atmosphere. Observations conducted during a storm in October 2015 in the Beaufort Sea have, for instance, revealed that storm-induced waves can lead to an increase of surface mixing and an associated heat entrainment from the upper ocean, resulting in large melt of pancake ice (Smith et al., 2018). Finally, waves transport momentum, and therefore when they are attenuated in the MIZ through reflection or dissipation, part of their momentum goes into sea ice. This process, called wave radiative stress (WRS; Longuet-Higgins and Stewart, 1962; Longuet-Higgins, 1977), acts as a force that pushes the sea ice in the direction of propagation of the attenuated waves. This force is a dominant term in the ice momentum balance in the Southern Ocean MIZ (Stopa et al., 2018b) and it may become more prominent in the Arctic in the future. In return, sea ice strongly attenuates waves propagating in the MIZ, either by dissipative processes (e.g. under-ice friction, inelastic flexure, floe-floe collisions) or conservative processes (e.g. scattering) (Squire, 2018).

Several recent efforts in the modelling community have been focused on the impact of sea ice on waves (Dumont et al., 2011; Williams et al., 2013; Montiel et al., 2016), leading to the development of wave models accounting for the presence of sea ice (Boutin et al., 2018). By prescribing sea ice conditions, these models are able to accurately reproduce the time and space variations of wave heights in sea ice retrieved from recent field observations (Kohout et al., 2014; Thomson et al., 2018; Cheng et al., 2017) and innovative processing of Synthetic Aperture Radar (SAR) satellite observations (Ardhuin et al., 2017). The good agreement with the observations also suggests a proper representation and quantification of wave attenuation and propagation in sea ice in these models (Ardhuin et al., 2016; Rogers et al., 2016; Ardhuin et al., 2018). Yet, in this case, sea ice conditions are only a forcing and thus not affected by waves. This means that these models cannot realistically represent the fate of the sea ice floes once broken by waves, as they do not account for advection, melting and refreezing processes. A first step towards the representation of wave-sea ice interactions was made by Williams et al. (2013) and Boutin et al. (2018), who included in their respective models a floe size distribution (FSD) that evolves depending on the sea state. However, considering only sea ice fragmentation is not sufficient to represent the full complexity of wave-sea ice interactions.

In parallel, progress has also been made regarding the inclusion of the effects of waves in coupled ocean-sea ice models. Using a very simple parameterization, Steele et al. (1989) and Perrie and Hu (1997) have investigated the effect of WRS on sea ice drift in the MIZ, only considering the attenuation of waves generated between the ice floes, and found a limited impact on the sea ice conditions. More recently, Williams et al. (2017) implemented a wave module in the semi-Lagrangian sea ice model neXtSIM (Rampal et al., 2016) and found that high wave conditions can cause a significant displacement of the sea ice edge. The implementation of FSDs in different sea ice models, as introduced by Zhang et al. (2015) and Horvat and Tziperman (2015) for instance, has also opened the way to the assessment of the potential enhancement of lateral melt by wave-induced ice fragmentation (Zhang et al., 2016; Bennetts et al., 2017; Roach et al., 2018b; Bateson et al., 2019), but the representation of waves remains too simple to simulate the full effect of waves on the evolution of sea ice.

In the present study, we go beyond simply forcing a wave model with sea ice properties, or conversely forcing a sea ice model by wave properties, by proposing a full coupling between a spectral wave model and a state-of-the-art sea ice model. The coupled framework allows us to investigate the interactions between waves and sea ice in the Arctic, and the impact that including these effects in a model has on the representation of the waves, ocean, and sea ice properties in the Arctic MIZ. We focus in particular on two aspects of these interactions: firstly the effect of including the WRS, computed by the wave model, in the sea ice model, and secondly the wave-induced sea ice fragmentation and its effects on lateral melt through the addition of a FSD in the sea ice model. The remainder of this paper is organized as follows. The different models and configurations used in this study are described in Section 2. Section 3 is devoted to the theoretical and practical implementation of the coupling between the two models. In section 4, we compare two pan-Arctic simulations: one for which the coupling between wave and sea ice is implemented, and one with the ocean-sea ice model run as stand-alone. Our objective is to quantify the dynamical and thermodynamical impacts of the coupling on the sea ice and ocean surface properties. A summary and conclusions are given in Section 5.

## 2    Methods

In this study we make use of the spectral wave model WAVEWATCH III® (hereafter WW3; The WAVEWATCH III® Development Group, 2016), building on the previous developments by Boutin et al. (2018) who included a FSD in WW3 as well as a representation of the different processes by which sea ice can affect the propagation and attenuation of waves in the MIZ. These processes are scattering (which redistributes the wave energy without dissipation), friction under sea ice (with a viscous and a turbulent part depending on the wave Reynolds number), and inelastic flexion. All these processes depend on sea ice thickness and concentration, and scattering and inelastic flexion also depend on floe size.

We also use the sea ice model LIM3 (Vancoppenolle et al., 2009; Rousset et al., 2015), in which a FSD is first implemented as described in Section 3.2. The model includes a standard Elasto-Visco-Plastic rheology (Hunke and Dukowicz, 1997), using the stress tensor formulation of Bouillon et al. (2013) adapted for the C-grid used in the model. The ice strength is determined

following Hibler III (1979), with the ice strength $P$ following $P = P^* h e^{C(1-c)}$, where $P^*$=20,000 N/m² and $C$=20 are empirical positive parameters, and $h$ is the cell-average sea ice thickness. The plastic failure threshold lies on an elliptical yield curve, with the eccentricity set to 2. The number of sub-time steps used to solve the momentum equation is set to 120. The two models are coupled through the coupler OASIS-MCT (Craig et al., 2017). Two configurations of different complexities are used in the following and briefly described in the remainder of this section.

## 2.1 Idealized configuration

In order to test and illustrate the effect of the coupling (Section 3), we make use of a simple idealized configuration (see Fig. 1), in which LIM3 is used in a stand-alone mode (without any ocean component). The configuration is a squared domain with $100 \times 100$ grid cells, with a resolution of 0.03° in both directions (corresponding roughly to 3 km). Both models are run on the same grid with the same time step set to 300s. The coupling time step is also set to 300s. The sea ice is forced solely by waves, without prescribing any wind or ocean currents. Following Boutin et al. (2018), the simulation starts at rest, with distributions of sea ice concentration (Fig. 1a) and thickness (Fig. 1c) set to represent roughly the conditions that can be found in the MIZ. Starting from the western border, the domain is free of ice over the first $\simeq$10 km, after which the ice concentration $c$ increases linearly from 0.4 to 1 about 90 km further eastward (longitude=0.84°E). Ice thickness also increases from west to east following $h_i = 2(0.1 + e^{-N_x/20})$, where $N_x$ is the number of grid cells in the $x$-direction starting from the western border of the ice-covered domain. Waves radiate from part of the western border of the domain, between 1.2 and 1.8° latitude, and propagate to the east. The wave spectrum used as forcing at the boundary is extracted at a point south of Svalbard from an Arctic hindcast performed with WW3 described by Stopa et al. (2016). It covers the period of May 2nd to 3rd, 2010, during which a storm occurred in this particular area (Collins et al., 2015). Here we rotate the spectrum so that the direction with the largest density of wave energy is lined up with our $x$-axis. Simulations start on April 30th, at 02:00 a.m., and the attenuation processes (scattering, bottom friction and inelastic flexion dissipation) use the same parameterization as in the reference simulation described by Ardhuin et al. (2018).

## 2.2 Pan-Arctic configuration

We also make use of the CREG025 configuration (Dupont et al., 2015; Lemieux et al., 2018), which is a regional extraction of the global ORCA025 configuration developed by the Drakkar consortium (Barnier et al., 2006). Although the coupling is solely between LIM3 and WW3, the configuration here also includes the ocean component of NEMO 3.6 (Madec, 2008). CREG025 encompasses the Arctic and parts of the North Atlantic down to 26°, and has 75 vertical levels and a nominal horizontal resolution of 1/4°($\simeq$ 12 km in the Arctic). Both NEMO-LIM3 and WW3 are run on the same grid. Initial conditions for the ocean are taken from the World Ocean Atlas 2009 climatology for temperature and salinity. The initial sea ice thickness and concentration are taken from a long ORCA025 simulation performed by the Drakkar Group. Along the lateral open boundaries, monthly climatological conditions (comprising sea surface height, 3-D velocities, temperature and salinity) are taken from the same ORCA025 simulation. Regarding the atmospheric forcing, we use the latest version of the Drakkar Forcing Set (DFS 5.2,

which is an updated version of the forcing set described in Brodeau et al., 2010). The choices regarding the parameterization of the wave-ice attenuation are following the ones made in the *REF* simulation by Ardhuin et al. (2018). The ice flexural strength has however been increased from 0.27 MPa to 0.6 MPa, which is the highest value used in Ardhuin et al. (2018). This choice makes sea ice harder to break, and has been made to compensate the fact that no lateral growth of sea ice is included in our coupled framework.

Three simulations are performed. First we run a simulation based solely on NEMO-LIM3 (referred to as NOT_CPL), covering the period from January 1st, 2002 to the end of 2010, in which the already existing lateral melt parameterization in LIM3 is activated. The first years of the simulations are allowing for the adjustment of the ocean and sea ice conditions and we only analyze results from August and September 2010. During that period, the sea ice extent reaches its annual minimum, providing 10 some fetch for the generation of waves, in particular in the Beaufort Sea. The model sea ice extent during the summer of 2010, and more generally the distribution of the sea ice concentration, compares reasonably well with satellite observations (not shown). Note that this period includes a drop in sea ice concentration in the central Arctic, found both in model results and in satellite observations, that has already been documented by Zhao et al. (2018) and attributed to an enhancement of ice divergence in this region in this particular year. This specific period has also been selected as it includes a few storm events, so 15 extreme wave conditions can be investigated. Another simulation (CPL) is initialized from NOT_CPL on August 1st 2010 and run until September 9th 2010. After this date, sea ice extent starts to increase again, and as our FSD distribution does not allow for the refreezing of sea ice floes, we cannot realistically represent the processes at play during this period.

Finally, we run a simulation over the same period, based solely on WW3 (referred to as WAVE), in which the wave model is forced by sea ice conditions from the NOT_CPL simulation. In order to allow for some spin up for the waves to develop and 20 break the ice, we exclude the first 3 days from the analysis. In the following, all the results are for the 37-day period between August 4th and September 9th 2010.

## 3  Implementation of the coupling between the wave and sea ice models.

The objective of this section is to present the theoretical background and the practical implementation of the coupling between LIM3 and WW3. Fig. 2 shows the principle of the coupling and the variables that are exchanged between the two models. 25 Briefly, LIM3 sends the sea ice thickness, concentration, and maximum floe size (estimated from the newly implemented FSD) to WW3. This maximum floe size, referred to as $D_{\max}$, represents the largest size of floes remaining after the fragmentation event. These quantities are used by WW3 in order to estimate the wave attenuation and wave-induced sea ice fragmentation. Note that, in general, we refer to sea ice thickness as the cell-average of the sea ice thickness distribution $g_h$ used as a state variable in LIM3. In the coupling, we actually exchange the ice-cover average sea ice thickness, although this choice does not 30 significantly affect our results. WW3 then returns the WRS to LIM3, as well as the updated maximum floe size if fragmentation has occurred. The occurrence of fragmentation is thus determined in the wave model, depending on the wave conditions (see Boutin et al., 2018). In LIM3, we assume wave-induced fragmentation results in a truncated power law defined for floe sizes

up to the maximum floe size estimated in WW3. LIM3 takes into account the WRS in its ice transport equation, and advects the sea ice and its FSD, which is defined as an areal distribution. The FSD in LIM3 is also used to estimate lateral melt.

In the following, we describe in more detail the modifications that have been carried out in LIM3 and WW3 in order to couple them, and how variables are exchanged between the two models. The coupling allows a new formulation for the sea ice lateral melt in LIM3 (section 3.3).

## 3.1 Wave Radiative Stress

Waves transport momentum, and when they are attenuated either by dissipation or reflection, this momentum is transferred to what has caused this attenuation (Longuet-Higgins, 1977). In the case of sea ice, this momentum loss thus acts as a stress that pushes sea ice in the direction of attenuated waves. Following the study of Williams et al. (2017), in which a WRS was implemented in neXtSIM, the WRS $\tau_{w,i}$ is computed as:

$$\tau_{w,i} = \rho_w g \int\limits_{0}^{\infty} \int\limits_{0}^{2\pi} \frac{-S_{\text{ice}}(\mathbf{x};\omega,\theta)}{\omega/k} (\cos\theta, \sin\theta) d\theta d\omega \tag{1}$$

where $\rho_w$ is the water density, $g$ is gravity, $\omega$, $\theta$ and $k$ are respectively the radial frequency, direction and wavenumber of waves and $S_{\text{ice}}(\mathbf{x};\omega,\theta)$ is the source term (negative by convention in WW3) corresponding to wave attenuation by sea ice at a given position.

Once estimated by WW3, the WRS is then sent to the sea ice model and added as an additional term in the momentum equation of LIM3 (Rousset et al., 2015):

$$m D_t \mathbf{u} = \nabla \cdot \boldsymbol{\sigma} + c(\tau_a + \tau_o) + \tau_{w,i} - m f \mathbf{k} \times \mathbf{u} - m g \nabla \eta, \tag{2}$$

in which $m$ is the total mass of ice and snow per unit of area, $\mathbf{u}$ is the ice velocity vector, $\boldsymbol{\sigma}$ is the internal stress tensor, $f$ is the Coriolis parameter, $\mathbf{k}$ is a unit vector pointing upwards, $\eta$ is the sea surface elevation, $c$ is the sea ice concentration, and $\tau_a$, $\tau_o$ are the atmospheric and oceanic stresses, respectively. In contrast to $\tau_a$ and $\tau_o$, $\tau_{w,i}$ does not need to be multiplied by $c$, as the wave attenuation estimation in WW3 (and hence the WRS) is already scaled by the sea ice concentration to account for the partial sea ice cover.

Fig. 1 illustrates the effect of the implementation of the WRS in our simple model. Here, the sea ice thermodynamics are switched off, so that we only simulate the effect of waves pushing sea ice. Under the action of waves, the sea ice edge shifts eastward, resulting in an increase of the sea ice concentration (panel b). As the sea ice near the sea ice edge is compacted, it creates a sharp gradient in sea ice concentration and thickness (panels b,d). When comparing panels (e) and (f), it is clear that wave attenuation also responds to this change of the sea ice properties: waves tend to penetrate further eastward when the sea ice edge retreats to the east, but are then attenuated faster in the compacted sea ice.

## 3.2 Floe size distribution and sea ice fragmentation

As mentioned earlier, waves can break sea ice and thus impact the sea ice floe size. It is therefore necessary to exchange parameters defining the FSD between the two models. A FSD has been previously implemented in WW3 by Boutin et al. (2018), and is used to estimate the wave attenuation due to inelastic flexure and scattering. Following the work by Toyota et al. (2011) and Dumont et al. (2011), we assume that the FSD in WW3 follows a truncated power law between a minimum floe size, $D_{\min}$ and a maximum floe size, $D_{\max}$. $D_{\min}$ corresponds to the minimum floe size that can be generated by waves and is of the order of $O(10m)$, while $D_{\max}$ depends on the local wave properties and is used to estimate the level of sea ice fragmentation. As $D_{\min}$ is assumed to be constant in WW3, the FSD in the wave model is thus only a function of $D_{\max}$. Ideally, WW3 would therefore send the value of $D_{\max}$ to the sea ice model, where it would be advected and updated due to the effects of thermodynamical and mechanical processes, and then sent back to WW3. Yet, $D_{\max}$ is not an area-conserved quantity, and therefore cannot be advected as a tracer (Williams et al., 2017). Instead, we thus choose to define a FSD in LIM3, from which a maximum floe size can be estimated and then sent to the wave model.

There is no FSD included in the standard version of LIM3. However, recent work by Zhang et al. (2015) and Horvat and Tziperman (2015) has proposed ways to implement a FSD in sea ice models, following what is done for the sea ice thickness distribution (which is a state variable of any multi-category sea ice model). In their study, Horvat and Tziperman (2015) use a joint thickness and floe size distribution in order to represent the evolution of sea ice floes affected by a great variety of processes not necessarily related to waves (i.e welding, refreezing, ridging...). The approach by Zhang et al. (2015) is simpler and computationally cheaper, as it assumes that all floes of a given size have the same ice thickness distribution, allowing the FSD to be treated independently from the sea ice thickness distribution. To do so, they hypothesize that the FSD mostly results from the fragmentation of large unbroken floes randomly yielding floes of any size smaller than the original ones. In this study, we choose to follow the simpler approach of Zhang et al. (2015), as we only consider the effects of wave-induced sea ice fragmentation and lateral melt on the FSD evolution, and our formulation of lateral melt does not depend on sea ice thickness (see section 3.3). Therefore, we can consider the distribution of sea ice thickness and floes independently, defined respectively as:

$$\int\limits_{h}^{h+dh} g_h(h)dh = \frac{1}{A}a_h(h, h+dh) \tag{3}$$

$$\int\limits_{D}^{D+dD} g_D(D)dD = \frac{1}{A}a_D(D, D+dD) \tag{4}$$

and respecting:

$$\int\limits_{0}^{\infty} g_h(h)dh = 1 \tag{5}$$

$$\int\limits_{0}^{\infty} g_D(D)dD = 1, \tag{6}$$

$h$ being the sea ice thickness and $D$ being the caliper diameter of the floes as defined by Rothrock and Thorndike (1984). In these definitions, $A$ is the total area considered, and $a_h$ and $a_D$ are the areas within $A$ covered by sea ice with thickness between $h$ and $h+dh$ and floes with diameters between $D$ and $D+dD$ respectively. The evolution of the FSD depends on sea ice advection, thermodynamics and mechanical processes, and is given by:

$$\frac{\partial g_D}{\partial t} = -\nabla \cdot (\mathbf{u} g_D) + \Phi_{th} + \Phi_m, \tag{7}$$

in which $\mathbf{u}$ corresponds to the sea ice velocity vector, $\Phi_{th}$ is a redistribution function of floe size due to thermodynamic processes (*i.e* lateral growth/melt), and $\Phi_m$ is a mechanical redistribution function associated with processes like fragmentation, lead opening, ridging, and rafting. In our implementation, the only processes affecting the FSD are lateral melt and sea ice fragmentation. Other processes driving the evolution of the sea ice concentration do not modify the shape of the FSD. From a
technical point of view, the FSD in LIM3 is implemented as an areal distribution divided into floe size categories. It is advected in the same way as other sea ice tracers like sea ice concentration or thickness.

In their sea ice model neXtSIM, Williams et al. (2017) have also implemented a FSD but in a different way than Zhang et al. (2015). Indeed, they assume that the FSD follows a truncated power-law between a minimum and maximum floe size, similar to the assumption made in WW3. Thus, in their model, the FSD always obeys a power-law with a constant exponent,
and only evolves when sea ice fragmentation results in a reduction of the maximum floe size. Here we combine the approaches of Zhang et al. (2015) and Williams et al. (2017) and implement a FSD in LIM3 that evolves following Eq.7, but that also undergoes a redistribution after each fragmentation event that makes it tend towards the power-law assumed in WW3. Thus, in contrast to Williams et al. (2017), we do not make any assumption about the shape of the FSD in general, but, just as they do, we assume that after being fragmented by waves, the FSD follows a power-law with a maximum floe size that depends on the
local sea state. Our implementation only differs from the one done by Zhang et al. (2015) in the way we have implemented the redistribution: instead of assuming that the broken sea ice is redistributed uniformly among the smaller floes after each fragmentation event (that actually quickly tends towards a power-law distribution with a varying exponent depending on the wave state in their study), we assume that wave-induced fragmentation results in a truncated power law with a constant exponent defined for floe sizes up to the value of $D_{\max}$ received from WW3, under the constraint of conservation of area. As a result,
the FSD resulting from fragmentation in LIM3 is the same as the one assumed in WW3, ensuring coherence between the two models. The exponent of the truncated power-law corresponds to a value of about -1.85 for the cumulative number of floes distribution, which is the distribution generally used to represent the FSD in the scientific literature (e.g Toyota et al., 2011;

Herman et al., 2018). This value originates from the parameterization of wave-induced fragmentation by Dumont et al. (2011),
and is the one used in WW3 by Boutin et al. (2018).This redistribution is instantaneous in LIM3, which we justify by the fact
that fragmentation is a violent process, that can completely change a FSD in time scales of a few minutes to a few hours (see
Collins et al., 2015, for the description of such an event). Note that the redistribution of the FSD due to fragmentation transfers
sea ice from large floes to smaller floes. Combined with lateral melt, a process that also reduces the floe size, the action of
waves in our model always reduces the floe size. The details of the mechanical redistribution function $\Phi_m$ are mostly following
what has been proposed by Zhang et al. (2015) and are given in appendix A.

Now that both models include a FSD, the two models can be coupled in order to represent the effect of the wave-induced sea
ice fragmentation, the occurrence of which in LIM3 is determined depending on information provided by WW3. As mentioned
earlier, sea ice fragmentation in WW3 is determined by local wave properties, and fragmentation events result in an update
of the maximum floe size $D_{\max}$ which controls the FSD. It is thus logical to define a similar parameter $D_{\max,\mathrm{LIM3}}$ from the
FSD of LIM3, that ideally equals the value of $D_{\max}$ in WW3. However, estimating $D_{\max,\mathrm{LIM3}}$ is not straightforward. Indeed,
our FSD implementation requires that $D_{\max,\mathrm{LIM3}}$ corresponds to the upper limit of the power law followed by the FSD in
both WW3 and LIM3, but also that $D_{\max,\mathrm{LIM3}}$ can evolve with the deviations of the FSD in LIM3 from this power-law under
the effects of sea ice advection and thermodynamics. Calling $g_{D,\mathrm{P.L}}$ the distribution corresponding to a FSD following the
assumed power-law, we thus define $D_{\max,\mathrm{LIM3}}$ as the greatest value of $D$ for which the following condition applies:

$$\int\limits_{D}^{\infty} g_D dD \geq k_{D_{\max}} g_{D,\mathrm{P.L}}, \tag{8}$$

in which $k_{D_{\max}}$ is an *ad hoc* parameter allowing the value of $D_{\max,\mathrm{LIM3}}$ to remain unchanged when the FSD slightly deviates
away from the assumed power-law (after lateral melt or advection for instance). Setting $k_{D_{\max}}$=1 results in an immediate re-
duction of Dmax if a negligible reduction in the proportion of large floes in the FSD after a fragmentation event (when lateral
melt occurs for instance). This can be problematic, as reducing the value of $D_{\max}$ during a fragmentation event allows for the
wave to propagate further, generating more fragmentation. Values of $k_{D_{\max}}$ between 0.5 and 0.8 allow for little variation in the
FSD while keeping the same value of $D_{\max}$, thus giving more weight to fragmentation than to other processes in the estimation
of $D_{\max}$.Overall the choice of $k_{D_{\max}}$ in this range does not significantly affect our results. In the following, $k_{D_{\max}}$ is set to 0.5.

Floes that have never been broken by waves have no physical reason to follow this truncated power-law. In practice, if we
consider a discrete number $N$ of floe size categories, the $N^{\mathrm{th}}$ category should represent these unbroken floes, with a different
condition to set the value of $D_{\max,\mathrm{LIM3}}$ to $D_N$ (the upper size limit of this category). We thus consider sea ice in a grid cell
as unbroken only if most of its floes belongs to this $N^{\mathrm{th}}$ category, so that $D_{\max,\mathrm{LIM3}} = D_N$ only if $g_N > 0.5c$, $g_N$ being the
value of the FSD function associated to the $N^{\mathrm{th}}$ category and $c$ the total sea ice concentration.

In all of our simulations, sea ice is initialized as unbroken everywhere, so that $g_N = c$, and $D_{\text{max,LIM3}} = D_N$. As soon as wave-induced fragmentation occurs, $D_{\text{max,LIM3}}$ is updated. To do so, the received value of $D_{\text{max}}$ is rounded up to the upper limit of the category it lies in. $D_{\text{max,LIM3}}$ is therefore slightly greater than the value received from WW3, with an error that depends on the width of its associated floe size category.

Tests with the simplified domain were also performed with different number and width of floe categories to investigate the sensitivity of the results to those parameters. This sensitivity remains very small as long as the widths of the categories are smaller than 10 m and the categories cover a range of floe sizes larger than 300 m. In the following, we use $N = 60$ floe size categories, that we define as follows:

- A first category corresponding to the sea ice floes that are already broken but cannot be broken anymore [$D_0 = 8$ m, $D_1 = 13$ m]. $D_0$ represents the smallest floe size possible in the model, and is set to 8 m in order to agree with the minimum floe size used in LIM3 to estimate lateral melt from the formula by Lüpkes et al. (2012). $D_0$ is also of the same order as the size of the smallest floes that can be generated by wave-induced fragmentation (Mellor, 1986) and therefore is an acceptable value for the lower limit $D_{\text{min}}$ that the truncated power-law is assumed to follow after wave-induced fragmentation.

- 58 categories for which $D_n - D_{n-1} = 5$ m, with $1 \leq n \leq N - 1$.

- A last category representing unbroken floes [$D_{N-1} = 298$ m, $D_N = 1000$ m]. This value of $D_N = 1000$ m is set as it is one order of magnitude higher than the floe size generated by waves (Toyota et al., 2011).

We evaluate the effect of this part of the coupling between WW3 and LIM3, as well as the robustness of the implementation of the FSD in LIM3, by looking at the same 2 simulations in our idealized configuration as presented in Fig. 1, and comparing an uncoupled WW3 simulation with a coupled WW3-LIM3 simulation (Fig. 3). Sea ice thermodynamics are switched off in the WW3-LIM3 simulation. In the uncoupled WW3 simulation, $D_{\text{max}}$ evolves depending on the sea state, but sea ice thickness and concentration are constant. In the WW3-LIM3 coupled simulation, sea ice properties are all evolving as sea ice is pushed by the WRS, and the FSD is advected in LIM3. The comparison between $D_{\text{max}}$ estimated from the WW3 simulation and $D_{\text{max,LIM3}}$ from the coupled simulation is shown in Fig. 3(a,b,c). The pattern of broken sea ice is broadly similar in the two simulations (a,b), despite the sea ice retreat due to the WRS in the coupled case. Differences in $D_{\text{max}}$ (Fig. 3c) follow the wave height differences already commented on in Fig. 1(e,f). Indeed, the retreat of the ice edge due to the WRS allows for waves to propagate further with less attenuation, thus involving more sea ice fragmentation and a lower maximum floe size close to the open ocean in the coupled simulation. Further east in the MIZ, the sea ice compacted by the WRS effect generates stronger wave attenuation, and thus less sea ice fragmentation and a larger maximum floe size when compared to the uncoupled simulation. Both effects partly compensate, so that the shift in the ice edge position has little effect on the spatial extent of broken ice. Fig. 3d shows the FSD at two locations in the domain. At both locations, the distribution of ice-covered area within the different categories agrees very well between LIM3 and the truncated power-law assumed in WW3. The area covered by floes of the smallest possible size in LIM3 is nevertheless greater than it would be if the FSD was exactly following the

truncated power-law. This is because floes that have been broken down into the smallest possible size do not contribute to the redistribution (see section A) and accumulate in this category since no lateral growth occurs. Note that a coupled simulation in which advection had been deactivated was also run to ensure that, in a case with unaffected initial sea ice properties, no significant discrepancies were noticeable for both significant wave height and maximum floe size between a coupled and an uncoupled simulation (not shown).

## 3.3 Lateral melt

A parameterization to account for the sea ice lateral melt is already implemented in LIM3. Its formulation follows Steele (1992):

$$\frac{dc}{dt} = -w_{\mathrm{lat}} \frac{\pi}{\alpha \langle D \rangle} c, \tag{9}$$

where $c$ is the sea ice concentration, $w_{\mathrm{lat}}$ is the lateral melt rate, which depends on the difference between sea ice and sea surface temperatures taken from Maykut and Perovich (1987), and $\alpha$ is a coefficient which varies with the floe geometry. By default, $\alpha = 0.66$, which is the average value of the non-circularity of floes obtained by Rothrock and Thorndike (1984). By default, $\langle D \rangle$, which represents the average floe size, is a function of the sea ice concentration obtained empirically from observational data by Lüpkes et al. (2012):

$$\langle D \rangle = D_{\min} \left( \frac{c*}{c* - c} \right)^{\beta} \tag{10}$$

where $\beta = 1$ and $c*$ is introduced to avoid a singularity at $c = 1$ and is defined as:

$$c* = \frac{1}{1 - (D_{\min}/D_{\min})^{1/\beta}} \tag{11}$$

This relationship finds a value of $\langle D \rangle$ that increases very little from its minimum value (set to $D_0$) as long as the sea ice concentration remains lower than $\simeq 0.6$ (see Fig. 3 from Lüpkes et al., 2012). In the following, we refer to this lateral melt parameterization as the parameterization of Lüpkes et al. (2012), although we acknowledge that the work of Lüpkes et al. (2012) only provides a relationship between the average floe size and the sea ice concentration.

In the case of our coupled model, we estimate a FSD, and thus it makes sense to implement a parameterization of the lateral melt that depends explicitly on the FSD rather than on sea ice concentration. Following the work by Horvat and Tziperman (2015) and Roach et al. (2018b), we estimate the lateral melt as:

$$\frac{dc}{dt} = \int_0^\infty \Phi_{th} dD = \int_{0+}^\infty -2w_{\mathrm{lat}} \left( -\frac{\partial g_D}{\partial D} + \frac{2}{D} g_D \right) dD \tag{12}$$

where $\Phi_{th}$ is the change in the FSD due to lateral melt (see Eq.7). Note that lateral melt for floes in the unbroken category is computed assuming that all the floes have a size D of 1000 m. Note also that Horvat and Tziperman (2015) and Roach et al. (2018b) represent the evolution of the floe radius, and therefore include in their equation a term they call $G_r$ representing $\frac{\partial r}{\partial t}$,

the rate at which the floe radius changes due to lateral growth/melt. In our case, we represent the evolution of the floe diameter, and the rate at which it changes due to lateral growth/melt is thus equal to $2\frac{\partial r}{\partial t}$ (hence the addition of a factor 2 in Eq.12).

We run two simulations, in which the lateral melt is either estimated from the formulation of Lüpkes et al. (2012), or by our new formulation, which accounts for the FSD that is determined by both the sea ice and the wave models (Fig. 4). Here we only activate the lateral melt, and turn off the basal and surface melt. The sea surface temperature is set constant to $T = 0.3$°C. Floe size categories are the same as in section 3.2. In the case of the Lüpkes et al. (2012) parameterization (Fig. 4a), lateral melt only depends on the sea ice concentration and thus follows its distribution. In the second case (Fig. 4b), lateral melt is highly constrained by both the distribution of the sea state and ice properties, and is only significant where the sea ice is broken. Melt rates are overall higher when estimated from the Lüpkes et al. (2012) parameterization, mostly due to the fact that the

average floe size in the uncoupled run is very close to $D_0$ for a wide range of concentrations. Unlike the parameterization that we propose here, the parameterization of Lüpkes et al. (2012) results in a significant lateral melt far from the ice edge, where sea ice is mostly compact and unbroken, which is likely unphysical.

We find that the results are also sensitive to the choice of $D_{\min}$, regardless of the parameterization used. In the case of our FSD, the sensitivity arises from the use of $D_{\min}$ to compute the average diameter of the smallest floe size category, which is

the category most affected by lateral melt. In order to quantify the sensitivity to the choice of $D_{\min}$, we run similar experiments to the one presented in Fig.4, varying this time the value of $D_{\min}$ (Fig. 5). When using the parameterization of Lüpkes et al. (2012) (Eq. 10 and 11), the volume of sea ice melted laterally roughly doubles when $D_{\min}$ is divided by 2 compared to the reference simulation (using $D_{\min}$ = 8 m). In the case of our FSD, the sensitivity is still large but greatly reduced, with an increase of only 20% in response to the same change of $D_{\min}$. Figure 5 also illustrates the strong differences in melted volume

between the two parameterizations, with much less lateral melt when computed using the FSD developed here.

## 4   Importance of wave-sea ice interactions

In this section we compare the three simulations performed with the CREG025 configuration described in Section 2.2, in order to quantify the impact of the coupling on wave, sea ice and ocean surface properties. To evaluate the impact of waves in the MIZ, we first need to define the MIZ in our model. Various criteria, relying either on sea ice concentration, floe size or the

region where waves impact the sea ice floe size, have been previously used to delimit the MIZ (see for instance Dumont et al., 2011; Strong and Rigor, 2013; Sutherland and Dumont, 2018). Here we take the following definition based on the maximum floe size: $0 < \langle D_{\max} \rangle < 700$ m, where $\langle D_{\max} \rangle$ is the average of the maximum floe size over the study period. Physically, it roughly corresponds to the region where sea ice has been broken during a time period that is long enough for the average maximum floe size to become under 1000 m (which is the limit between the broken and unbroken ice). Note that our results

are not dependent on the definition of the MIZ.

## 4.1 Effect of the coupling at the pan-Arctic scale

### 4.1.1 Impact on the wave properties

First, we examine the differences between the CPL and WAVE simulations, corresponding respectively to the coupled WW3-LIM3 run and a run performed with WW3 in stand-alone mode forced with sea ice properties from NOT_CPL. When looking at the differences in significant wave height $H_s$ (Fig. 6b), we find that they are small, not exceeding $\simeq$ 15 cm on average. Moreover, the two runs exhibit similar patterns of $D_{\max}$, indicating that the wave-induced fragmentation is similar in the two simulations (Fig. 6d). Locally, in the Barents and Greenland seas for instance, the differences in $D_{\max}$ can be significant, due to the specific ice drift conditions in these regions. Indeed, the overall southward drift of sea ice tends to bring unbroken sea ice from the central Arctic to regions where sea ice is broken up, increasing $D_{\max}$ in the CPL simulation. The signs of the differences in $H_s$ and $D_{\max}$ vary regionally. This might be due to the differences in sea ice concentration and thickness, as the wave attenuation in sea ice is very sensitive to sea ice properties (see for instance Ardhuin et al., 2018). Certainly, the pattern of the differences in $H_s$ between the CPL and WAVE runs is consistent with the differences in sea ice concentration and thickness between the CPL and the NOT_CPL simulations (Fig. 7), with higher waves found in regions where ice is less concentrated and thinner.

### 4.1.2 Impact on the sea ice and sea surface properties

We now focus on the effect of adding a wave component on the sea ice properties, by comparing results from the CPL and NOT_CPL simulations. Fig. 7 shows the pan-Arctic distribution of the sea ice thickness and concentration averaged over the 37 days considered in the CPL simulation, as well as the differences with the NOT_CPL simulation. These differences are concentrated in the vicinity of the ice edge and exhibit different signs depending on the location. Positive and negative anomalies tend to compensate, resulting in weak overall differences in sea ice extent and volume when averaging over the full Arctic Basin. If we only consider the MIZ, the sea ice volume and area decrease by about 3% and 2%, respectively, between CPL and NOT_CPL ( Fig. 8b). Locally, however, these variations can be much larger. In the MIZ of the Beaufort Sea for instance, the relative changes can be as high as 10% for grid cell-average thickness.

There are also differences in sea surface properties between the two simulations (Fig. 9), with average increases in sea surface temperature (SST) and salinity (SSS) in the MIZ as high as 0.5°C and 0.8 psu locally, respectively. It is worth noting that, in contrast to the sea ice properties, the sign of the differences in SST and SSS tends to be positive, i.e. warmer and saltier in the CPL experiment compared to the NOT_CPL one.

### 4.1.3 Thermodynamical effect of the coupling

Given that there is no coupling between the ocean and the wave components, the difference in sea surface properties must arise from variations in sea ice conditions, and in particular the sea ice melt, and we now investigate this further. Fig. 10(a,b) shows the total sea ice volume loss from lateral melt during the study period in the CPL run as well as the difference between this and the same quantity from the NOT_CPL run. The sea ice volume melted by lateral melt shows very similar spatial patterns in the two simulations, although it is estimated from two very different parameterizations (Eq. 9 and Eq. 12), although lateral

melt estimated by the parameterization from Lüpkes et al. (2012) tends to be lower in CPL. The difference is substantial, the sea ice volume melted in the MIZ in CPL being 30% lower than in NOT_CPL (Fig. 8a). Another signal is found in the central Arctic, while no lateral melt occurs in this region in CPL. This signal actually arises from the combination of the drop in sea ice concentration that happens in the region in August 2010 (Zhao et al., 2018), and the use of the parameterization by Lüpkes et al. (2012) to estimate floe size and resulting lateral melt in NOT_CPL. Indeed, lower sea ice concentration values corresponds to estimated average floe sizes below 100 m when estimated from the parameterization by Lüpkes et al. (2012), and thus some lateral melt is triggered. In contrast, the absence of waves in the middle of the sea ice pack in CPL leaves sea ice unbroken in this region, preventing lateral melt from occurring. An average floe size of $\simeq 100$ m in the middle of the pack seems somewhat unrealistic, and highlights the limitations of estimating the floe size using Lüpkes et al. (2012) parameterization in pan-Arctic

configurations. Absence of lateral melt in the central Arctic in CPL explains the excess in sea ice concentration when compared to the uncoupled simulation (Fig. 7b). Moreover, the fact that differences in lateral melt between the two simulations are mostly negative means that it cannot explain the regional patterns found in the distribution of sea ice property differences.

Fig. 10(c,d) shows the differences in bottom and total ice melt, between the CPL and NOT_CPL simulations. The spatial dis-

tributions of the differences in bottom and total ice melt are very similar, meaning that the variations in bottom melt dominate the differences in sea ice melt between CPL and NOT_CPL, although the bottom melt is computed the same way in the two simulations. This result is confirmed by rerunning a coupled and an uncoupled simulation of NEMO-LIM3 while de-activating lateral melt (not shown), which yields differences in total melt distribution almost identical to the ones presented in Fig. 10(c,d).

The total melted sea ice volume, once integrated over the MIZ, increases by 3% between CPL and NOT_CPL, mainly due to the larger volume of sea ice melted laterally in NOT_CPL (Fig. 8a). In parallel, bottom melt slightly decreases by $\simeq 1\%$ between these two simulations. This result masks the fact that the regional differences of total melt are dominated by bottom melt. An explanation is that bottom and lateral melt both depend on the available heat in the surface layer, either directly for bottom melt, or indirectly through lateral melt that depends on the SST. If lateral melt occurs, it removes heat from the surface

layer, therefore reducing the bottom melt capacity. Conversely, if this heat is not used for lateral melt, it remains available for bottom melt. The overall decrease of bottom melt in the MIZ between CPL and NOT_CPL visible in Fig. 8a therefore mostly results from the compensation of the increase of lateral melt due to the change of parameterization, as can be seen in Figs. 10b and 10c. This compensation mechanism has been reported before by (Tsamados et al., 2015; Roach et al., 2018a), and also by

Roach et al. (2018b) and Bateson et al. (2019) who compare two runs of the sea ice model CICE, one with the standard lateral
melt parameterization using a constant floe size of 300 m and one using a FSD (allowing floes smaller than 300 m). In our
case, this compensation is strong enough, and completely de-activating lateral melt in both runs has a negligible effect on the
quantity of melted ice in our simulations.

### 4.1.4 Dynamical effect of the coupling

The differences in lateral melt between the CPL and the NOT_CPL runs cannot explain the differences in sea ice and sea surface
properties seen in Figs. 7 and 9. We thus investigate the impact of the WRS on the sea ice conditions and melt. Fig. 10(e,f)
show the mean directions of the wind stress and the WRS in the CPL simulation and the ratio of WRS magnitude to wind
stress respectively. This ratio is generally low, not exceeding 15% of the wind stress in the eastern Barents Sea, where the WRS
reaches its highest magnitude. This is much smaller than the values retrieved from satellite observations in the Southern Ocean,
where the wind stress and the WRS can be of comparable magnitude (Stopa et al., 2018a). It is also worth noting that the
regions where this relative importance of the WRS compared to the wind is large do not always coincide with regions where
differences in sea ice properties are significant (Fig. 7). In the Beaufort Sea for instance, there is substantially less sea ice melt
in the CPL simulation than in the NOT_CPL one, although the ratios of WRS over the wind stress are only of the order of a
few percent (Fig. 10f). The opposite situation is visible in the Barents Sea, where the high relative influence of the WRS does
not result in a significant increase of the sea ice melt when the effect of the waves is included. Therefore, there is no direct
relationship between the intensity of the WRS and the differences in sea ice and sea surface properties between the coupled
and uncoupled simulations. In the Southern Ocean, Stopa et al. (2018a) found that the orientation of the WRS, which tends to
be orthogonal to the sea ice edge, might explain why WRS is as important as the wind (that tends to vary in direction much
more over time) in determining the position of the sea ice edge. Similarly, here, we found that the WRS is very often orientated
orthogonally to the ice edge, towards packed ice. This is due to the fact that the longer waves encounter sea ice on their path, the
more they are attenuated. The direction of propagating waves at a given point in sea ice is then generally imposed by the waves
that have travelled the shortest distance in sea ice. This is particularly visible in some parts of the Greenland and Kara seas,
where wind and wave stresses have opposite directions on average. In the Chukchi and the eastern Beaufort seas, the WRS is
orthogonal to the wind stress. In contrast, in the Laptev Sea, the directions of the WRS and the wind stress roughly align, and
thus work together in setting the position of the sea ice edge in the CPL run. However, at the pan-Arctic scale, there is no clear
relationship between the WRS direction and the differences in sea ice melt induced by the WRS in the CPL simulation.

The primary effect of the WRS is to push sea ice, modifying the intensity and the direction of the sea ice drift. This impact
is significant in the MIZ, where the average sea ice drift velocity increases by $\simeq 9\%$ between the CPL and the NOT_CPL runs
(Fig. 8b). This overall increase of the sea ice velocity can be explained by the fact that both WRS and sea ice drift have a
dependency on wind direction. As was the case for sea ice thickness and concentration, the distribution of the differences in
sea ice drift velocity between the two simulations varies strongly depending on the region considered (not shown), but exhibits
no clear relationship at the pan-Arctic scale that could explain the differences in sea ice melt induced by the WRS.

In the following we investigate in further detail the wave-sea ice interactions in two regions during storms. Indeed, although the differences between the CPL and NOT_CPL run at the pan-Arctic scale remain small, it is clear that the way the waves can influence the sea ice and the ocean surface would depend on the local properties of the waves, wind, sea ice and ocean surface.

## 4.2 Regional impacts of wave-sea ice interactions during storm events

### 4.2.1 Case 1: Storm in the Beaufort Sea (16-17 August 2010)

We first focus on a storm event that occurred near the MIZ in the Beaufort Sea on 16-17 August 2010 (Figs. 11(a,b,c) and 12(a,e)). During the storm, waves and winds are oriented towards the north-west on the west side of the domain, but towards the west on the east side. Wave height and wind speed are reaching up to 3 m and 12 m/s (Fig. 11a,b), respectively, while they do not exceed 1 m and 7 m/s during the 3 days preceding the storm (not shown). Before the event, the south Beaufort Sea is ice-free, and the position of the sea ice edge (defined at the 15% sea ice concentration contour) is highly irregular, with the presence of an ice tongue centered around 72°N and 155°W, that is exposed upwind on its eastern side but downwind on its western side during the storm. This sea ice tongue is composed of relatively thick ice ($\geq$ 1m). During the storm, sea ice breaks all over the ice tongue in the western part of the domain, but not further than $\simeq$40 km after the sea ice edge. Both the waves and the wind stresses push the ice to the west (Fig. 11b,c), accelerating the drift that is directed north-west (Fig. 12a,c), as was already the case before the storm (not shown). The wave action is particularly effective at the location of the sea ice tongue, where the WRS has an amplitude comparable to the wind stress over sea ice (Fig. 11c). As a consequence, the sea ice drift is substantially accelerated (Fig. 12c). The effect of the waves results in large changes of the sea ice thickness pattern (when comparing the CPL and NOT_CPL runs), with a decrease on the eastern part of the tongue but an increase on the western part (Fig. 12g). Outside of the sea ice tongue, the differences between the simulations are very small, likely because of the sharp sea ice thickness gradient opposing internal resistance to deformation (Fig. 12e), and the relative small effect of the WRS compared to the wind stress (Fig. 11c).

The differences in sea ice properties around the sea ice tongue between the two runs also result in changes in SST and SSS, with increases around 1°C and 1 psu, respectively, on the eastern side of the sea ice tongue, and a decrease of roughly the same magnitude on the western side (Fig. 13c,g). These differences arise from changes in sea ice melt, as differences in the total heat flux at the sea surface (Fig. 14a) are largely determined by bottom melt (Fig. 14b), the lateral melt contribution being one order of magnitude lower in this case. On the eastern side of the sea ice tongue, waves tend to push the sea ice away from the edge in the CPL run, and thus away from surface waters with warmer SST, resulting in a smaller amount of heat in the surface layer available for bottom melt. As the sea ice melt decreases, it also reduces the amount of freshwater received by the ocean surface, resulting in larger SSS. On the western side of the south end of the ice tongue, where the sea ice is thicker in the CPL run than in the NOT_CPL one, the opposite effect happens, explaining the lower SST and SSS values. One should note that the effects of this storm are particularly strong, due to the specific conditions before the storm, with warm waters brought very

close to the sea ice edge during the storm (not shown) .

In our model, bottom melt arises from heat fluxes determined by two distinct processes: (i) a conductive heat flux, the intensity of which is controlled by the difference between sea ice temperature and SST, and (ii) a turbulent heat flux in the surface layer, which depends on both the SST and the shear between the sea ice and the sea surface currents. The inclusion of the WRS could in principle affect the turbulent heat flux through its effect on the sea ice drift, but it is not the case here, suggesting that the deficit of sea ice melt on the eastern side of the sea ice tongue in the CPL run is therefore due to the combination of colder

SST and sea ice reduction.

### 4.2.2   Case 2: Storm in the Barents Sea (16-17 August 2010)

The storm that we just examined in the Beaufort Sea occurred on the same date as a second and stronger storm in the Barents Sea, with wave heights up to 5 m and south-westward winds reaching $\simeq$15m/s on average over the two days (bottom panels of Fig. 11d,e). In the CPL run, waves fragments sea ice over a very large area (Fig. 12f). Similarly to what we see in the Beaufort Sea, the mean direction of propagation of the waves aligns with the direction of the wind over the ice-free ocean, and is rotated orthogonally to the gradient in sea ice thickness once in the sea ice pack (Fig. 11d). The transition is however much smoother

here than in the Beaufort Sea as the gradient is much weaker (Fig. 12f). In the CPL run, sea ice is drifting southward (Fig. 12b), with a slight deviation from the wind direction, and speeds twice as large as in the Beaufort Sea, due to stronger winds and thinner and less concentrated sea ice.

    In contrast to the effects of the storm in the Beaufort Sea, the WRS in the CPL run reaches large values (Fig. 11f). Indeed,

the strong storm generates high waves, inducing a WRS as large as the wind stress close to the sea ice edge where most of the attenuation takes place, although the WRS does not align with the direction of the wave propagation in ice. This is due to the low sea ice concentration in this region that allows for wave generation over a large region, even if partially ice-covered. The attenuation of these short in-ice generated waves dominates the WRS that is therefore aligned with the wind direction, thus accelerating the ice drift, especially close to the ice edge (Fig. 12d).

    The differences in sea ice drift between the CPL and the NOT_CPL runs also result in differences in bottom melt (Fig. 14d), and more specifically in the part associated with the turbulent heat flux (not shown). This increase of the turbulent heat flux, which occurs in the Barents Sea but not in the Beaufort Sea, can be explained by the larger ice drift velocities driven by the WRS, which intensify the shear between the sea ice and the ocean, and therefore the turbulence in the surface mixed layer.

The differences in sea ice drift between the two runs also result in changes of the conductive heat flux. However, in the Barents Sea, the sea ice thickness and concentrations are lower than in the Beaufort Sea while the sea ice temperature is higher overall (not shown). This results in only moderate differences of the conductive heat flux between the CPL and NOT_CPL runs.

The differences in SST and SSS exhibit similar patterns to the differences in heat flux (Fig. 13d,h and Fig. 14c), but the magnitude of the differences are much weaker than in the Beaufort Sea, not exceeding a few tenths of $^{\circ}$C and psu for SST and SSS respectively. These small differences can be explained by two causes: (i) the small differences of sea ice properties between the two simulations result in small changes in melt, and (ii) the initial state before the storm is also different with higher SST and SSS in CPL (not shown). This difference in the initial state can be related to previous wave and wind conditions: low wind speeds are not sufficient to generate waves in the MIZ, implying that the WRS must be directed northward in the same direction as the propagating waves. It therefore compacts the sea ice edge, and thus reduces sea ice melt in the MIZ in the CPL run. As seen in the Beaufort Sea case, this in turn leads to higher SST and SSS values in the vicinity of the ice edge.

### 4.2.3 What determines the impact of the waves?

From these two particular cases we suggest a generalization of the mechanisms by which the waves can impact the sea ice and ocean properties in the MIZ. It is based on a simple principle: if sea ice is moved towards warmer water, it tends to melt more, and *vice versa*. The direction of the WRS compared to the orientation of the sea ice edge is thus fundamental if we are to understand the impact of the waves. In compact sea ice, waves are quickly attenuated and the direction of the WRS is generally towards the packed ice, thus impeding part of the sea ice melt and increasing the SST and SSS (Fig. 9). In regions where the sea ice is less concentrated and thinner, waves can be generated locally, so that the WRS aligns with the wind, whose direction determines the impact of the WRS (enhanced melt for off-ice wind and reduced melt for on-ice wind). Another key factor determining the impact of the WRS on sea ice is the internal stress of sea ice (a.k.a the rheology; see Eq.2). The impact of the WRS is larger in regions of the MIZ where the sea ice is thin and has low concentration, as the internal stress tends to be negligible (Hibler III, 1979), making the sea ice easier to deform and to drift freely. Close to the sea ice edge in the Barents Sea for instance, the WRS in storm-induced high wave conditions can be larger than the wind stress, strongly accelerating the sea ice drift towards the open ocean, which also results in an increase of the ice-ocean shear, enhancing the turbulent heat flux under sea ice and thus the sea ice melt.

## 5 Discussion and conclusion

The goal of this study was to examine the wave-sea ice interactions in the MIZ of the Arctic Ocean during the melt season, as these processes are thought to be important for determining the sea ice conditions but are not accounted for in the state-of-the-art sea ice models. To that aim, we have developed a model framework, coupling the wave model WW3 with a modified version of the ocean/sea ice model NEMO-LIM3. The coupled model was then used to examine two aspects of the wave-sea ice interactions: (i) the impact of the WRS on the sea ice drift in the MIZ, and (ii) the effects of using the wave-induced sea ice fragmentation to estimate lateral melt. The WRS tends to compact the ice edge and thus reduces the total sea ice melt in the MIZ. Yet, its overall impact on the MIZ sea ice area and volume remains limited (Fig. 8b). However, it has a visible impact on sea ice drift velocity, accelerating it by $\simeq$9%. Compared to the use of the parameterization of Lüpkes et al. (2012) to estimate

the floe size used in lateral melt, our parameterization strongly reduces the amount of sea ice melted laterally. It is however
mostly compensated by an increase of bottom melt, similar to what was found by Bateson et al. (2019). As a result, the effects
on sea ice and sea surface properties can be locally substantial, and even more substantial during storms, as illustrated by the
case studies in the Beaufort and Barents seas. As the storminess in the Arctic region is expected to increase in the future (Day
et al., 2018; Day and Hodges, 2018), generating higher and more energetic waves more frequently (Khon et al., 2014), the
wave-sea ice interactions might become a dominant signal controlling the dynamics of the MIZ.

In the MIZ, waves push sea ice as they are attenuated, locally modifying the position of the sea ice edge through a mod-
ulation of the magnitude and timing of the sea ice melt, which results in significant changes of the SST and SSS. Although
the impact at the pan-Arctic scale remains limited, the case studies of storms in the Barents and Beaufort seas show how this
modulation can be locally and intermittently important. Results from our simple configuration have also revealed that the WRS
could strongly modulate the position of the sea ice edge. Yet, except very locally in response to strong storms, the position of
the pan-Arctic sea ice edge simulated by our realistic configuration appears to be insensitive to the effect of the wave. This is
likely because the position of the sea ice edge in a ocean-sea ice model is primarily determined by the atmospheric forcing and
the bulk formulae, and is in particular strongly tied to the position of the sea ice edge in the atmospheric reanalysis (Chevallier
et al., 2017). The effects of the waves on sea ice simulated by our coupled model are likely underestimated, and should be
re-assessed in future studies based on a fully coupled model that includes an atmospheric component.

We have also tested two parameterizations of the lateral melt, based either on wave-induced fragmentation information or
solely on a scaling between the size of the floes and the sea ice concentration, following Lüpkes et al. (2012). We first ac-
knowledge that the effect of our lateral melt parameterization depends strongly on the FSD, and hence on the choices and
assumptions made regarding its implementation. For instance, our redistribution scheme associated with sea ice fragmentation
assumes that successive fragmentation events lead to a power-law FSD. This assumption is made based on the observations
analyzed by Toyota et al. (2011), that only sample a small area in time and space, and their findings may not be applicable
globally. More generally, Roach et al. (2018b) recommend avoiding forcing the shape of the distribution, as the analysis of
observations have revealed that FSDs do not always follow power-law distributions (e.g. Inoue et al., 2004). They foster the
use of alternative approaches, such as the one developed by Horvat and Tziperman (2015). However, results from laboratory
experiments focusing on the fragmentation of sea ice by waves by Herman et al. (2018) indeed suggest power-law distributions
for the smallest floe sizes generated, similarly to what was found by Toyota et al. (2011) for the small floes regime. This jus-
tifies the choice made in the present study. More generally, large uncertainty remains regarding the key parameters governing
the FSD redistribution (coming from waves or sea ice properties), and more dedicated observations will be needed in the future
to better constrain FSD in models.

Regardless of the choices made for the implementation of the FSD, the effect of the lateral melt for both formulations re-
mains limited as any change of lateral melt tends to be compensated by an opposite change of bottom melt. The effect might

however become more important if longer simulations were performed. Indeed, Zhang et al. (2016) found that, over a year, the lateral melt could significantly affect the sea ice thickness. In their case, a FSD-based parameterization was used (similar to the one we introduced in our coupled model), but the effect of the wave-induced fragmentation on the FSD was only crudely parameterized, resulting most likely in an overestimation of lateral melt in the central Arctic (as this is the case when using the parameterization of Lüpkes et al., 2012). Adding a FSD in their sea ice model, Roach et al. (2018b) found a large impact on sea ice concentration in the MIZ and sea ice thickness everywhere in the Arctic after 20 years of simulation, and suggested that the differences found in the central Arctic result from a redistribution of the heat used for lateral melt instead of bottom melt, similar to what happens in our model over a shorter timescale. One should also remember that the studies of Zhang et al. (2016) and Roach et al. (2018b) aimed to represent the evolution of floes with sizes ranging from a few cm to roughly 1 km on long time scales, whereas we focus on the important processes for wave-sea ice interactions and make the assumption that unbroken floes have a uniform floe size set to 1000 m. Therefore we do not expect any impact of the lateral melt in regions that are not impacted by waves. Note also that we evaluate the impact of changing the lateral melt parameterization by comparing two simulations for which lateral melt depends on a varying floe size, either deduced from the FSD or estimated from the sea ice concentration using the parameterization suggested in Lüpkes et al. (2012). It differs from Zhang et al. (2016) who compare their FSD-model with a reference run without lateral melt, and from Roach et al. (2018b) who use a constant floe size of 300 m in their lateral melt parameterization. This might partly explain the discrepancies between our respective conclusions.

Among the wave-sea ice interaction processes considered in this study, we find that the dynamical effect of the waves (the WRS) has a larger impact on sea ice conditions and sea surface properties than the modulation of lateral melt by sea ice fragmentation. Our simulations were however limited to only a few weeks during the melting season and it is unclear if the result would hold if longer timescales were considered. In order to answer this question, we would need to implement a parameterization that accounts for the refreezing of floes, through lateral growth and welding. A first parameterization of this kind has been very recently developed by Roach et al. (2018b). We also anticipate that running a simulation over longer time periods would highlight new impacts of the WRS. Indeed, observations have revealed that heat stored during the melt season below the mixed layer can significantly affect sea ice growth the following year (Jackson et al., 2010; Timmermans, 2015). In regions where the WRS contributes to reducing the ice melt, an excess of summer heat could likely accumulate under the mixed layer, possibly modulating the future evolution of the sea ice melt and growth. Recently, Smith et al. (2018), for instance, observed that a large amount of heat stored under the mixed layer could be released to melt sea ice during a storm. The significant changes of SST and SSS found locally over 37 days also highlight that wave-sea ice interactions should be considered when trying to forecast the Arctic sea ice conditions on short timescales (up to a few weeks), as these surface ocean changes can greatly affect melting and refreezing conditions.

The coupling developed in this study marks a valuable new step towards an improved representation of waves and sea ice interactions in models, which might improve the representation of the dynamics in the MIZ. Yet, our coupling relies on a number of assumptions, which are most likely leading to an underestimation of the impact of the waves on the ocean and sea ice

conditions. For instance, in our coupling, the sea ice rheology is unaffected by fragmentation, which is unlikely to be the case
in reality (McPhee, 1980). Moreover, the sea ice model used here does not retain any memory of the past sea ice conditions, while waves would most likely have a different effect on sea ice that has been previously broken (Langhorne et al., 1998). Developing a similar coupling using a model that considers a state variable accounting for the previous sea ice conditions (such as in the neXtSIM sea ice model (Rampal et al., 2016; Williams et al., 2017)) would probably reveal new mechanisms via which waves can modulate the ocean and sea ice conditions in the MIZ.

Finally, the coupling we have developed here is also only considering the interactions between waves and sea ice, without any direct coupling with the ocean and the atmosphere. Yet, we know that wave dissipation would also likely impact the mixed
layer, by enhancing turbulence (Couvelard et al., Submitted), and eventually modulate the rate of sea ice melt and formation (Martin and Kauffman, 1981; Rainville et al., 2011; Lee et al., 2012; Smith et al., 2018). Similarly, the effect of the waves is probably damped due to the lack of feedbacks with the atmosphere (Khon et al., 2014). Future coupling should include some of these features in order to fully capture the complexity of the MIZ dynamics.

*Code and data availability.*    The Drakkar Forcing Set is available at https://ige-meom-opendap.univ-grenoble-alpes.fr/thredds/catalog/meomopendap/
extract/FORCING_ATMOSPHERIQUE/DFS5.2/ALL/catalog.html. The wave-sea ice coupling routines developed in WW3 are included in the publicly available latest release of the code (v6.07) at https://github.com/NOAA-EMC/WW3. The modified routines of NEMO-LIM3 allowing for the wave-sea ice coupling are available at https://zenodo.org/badge/latestdoi/232142979. Configuration files are available upon request to claude.talandier@ifremer.fr or camille.lique@ifremer.fr. Model outputs are available upon request to guillaume.boutin@nersc.no.

## Appendix A: Floe size redistribution in the sea ice model LIM3

Here we provide the details of the calculation and implementation of the FSD, and in particular of the mechanical redistribution function $\Phi_m$ that accounts for processes such as sea ice fragmentation, lead opening, ridging, and rafting. Following Zhang et al. (2015), $\Phi_m$ can be divided into 3 terms as $\Phi_m = \Phi_o + \Phi_r + \Phi_f$ where $\Phi_o$ represents the creation of open water, $\Phi_r$ represents sea ice ridging and rafting, and $\Phi_f$ represents the wave-induced floe fragmentation. Here we compute $\Phi_o$ and $\Phi_r$ in a similar way to Zhang et al. (2015), assuming that all the floes of different sizes have the same ice thickness distribution, so
that changes in sea ice concentration due to open water creation or ridging affect all floes equally. As a result, the shape of the FSD and its evolution are independent from these two terms.

Assuming that, in a given grid cell, sea ice fragmentation does not induce any change of the sea ice concentration, $\Phi_f$ can be written as (Zhang et al., 2015):

$$\Phi_f = -Q(D)g_D(D) + \int_0^\infty Q(D')\beta(D',D)g_D(D')dD' \tag{A1}$$

where D is the floe size, $Q(D)$ is a redistribution probability function characterizing which floes are going to be broken depending on their size, and $\beta(D',D)$ is a redistribution factor quantifying the fraction of sea ice concentration transferred from one floe size to another as fragmentation occurs. $\Phi_f$ is thus used to transfer sea ice concentration from large floes to smaller floes. To ensure the conservation of sea ice area during fragmentation, $\beta$ must respect (Zhang et al., 2015):

$$\int\limits_0^\infty \beta(D',D)dD = 1 \tag{A2}$$

In the absence of a wave model to simulate the sea state, Zhang et al. (2015) defined $\beta$ so that it uniformly redistributes the sea ice concentration of the large broken floes into the smaller floe size categories of the FSD. Their redistribution probability function $Q(D)$ thus assumes that a constant fraction of the sea ice cover is broken by waves during each fragmentation event. Their definition of $Q(D)$ also ensures that larger floes contribute more to the redistribution than smaller floes.

In our coupled model, sea ice fragmentation is initially computed by WW3 (for details see Boutin et al., 2018), and accounts for the sea state variability. In WW3, the FSD resulting from wave-induced fragmentation is assumed to follow a truncated power-law between a minimum ($D_{\min}$) and a maximum ($D_{\max}$) floe size. For consistency, the FSD in LIM3 after a given fragmentation event must follow the same power-law, defined for D taken in $[D_{\min}\ D_{\max}]$ such as, for a given floe size $D_*$ also taken in $[D_{\min}\ D_{\max}]$:

$$P(D > D_*) = KD_*^{-\gamma}, K \in \mathbb{R} \tag{A3}$$

$$p(D) = -K\gamma D^{-\gamma-1} \tag{A4}$$

where $P(D > D_*)$ is the probability of having $D > D_*$, and $p(D)$ is the associated probability density. In WW3, a fragmentation event occurs if two conditions are met: (i) waves with a wavelength $\lambda$ apply a strain on sea ice greater than a given threshold, and (ii) $\lambda/2$, which is assumed to be the value of the new maximum floe size, is lower than the current $D_{\max}$ value in the wave model (Dumont et al., 2011). Therefore, a fragmentation event in WW3 corresponds to a decrease of $D_{\max}$.

As detailed in section 3.2, we define a maximum floe size in LIM3, $D_{\max,\text{LIM3}}$, that is compared to the value of the maximum floe size received from WW3, $D_{\max,\text{WW3}}$. Initially, ice is unbroken and $D_{\max,\text{LIM3}} = D_{\max,\text{WW3}}$. If fragmentation has occurred in WW3, then we have $D_{\max,\text{WW3}} < D_{\max,\text{LIM3}}$. In this case, $D_{\max,\text{LIM3}}$ must be updated to the $D_{\max,\text{WW3}}$ value, and $\Phi_f$ must be computed so that it forces the FSD in LIM3 to match the FSD assumed in WW3.

In practice, in LIM3, we define a given number $N$ of floe size categories, such that each floe size category $n \in [0,N]$ represents the floes with sizes in $[D_{n-1}, D_n]$. $D_0$ and $D_N$ are the minimum and the maximum floe size possible in the model, respectively. $D_N$ aims to represent floes that have not been broken by the waves. In WW3, the size of unbroken floes is set to 1000 m, and we thus also set $D_N$=1000m for consistency. Regarding the minimum floe size resulting from wave induced fragmentation, we set $D_{\min}$ to 8 m, which is the value of the minimum floe size used in the parameterization of lateral melt implemented in LIM3. This value is close to choices made in previous studies (see Williams et al., 2013; Bennetts et al., 2017).

If fragmentation occurs, the update of $D_{\mathrm{max,LIM3}}$ is done as follows:

$$\begin{cases} D_{n^*-1} < D_{\mathrm{max,WW3}} \leq D_{n^*} \\ D_{\mathrm{max,LIM3}} = D_{n^*}. \end{cases} \tag{A5}$$

Here $n^*$ is the index of the floe size category in which the maximum floe size received from the wave model lies in. $D_{\mathrm{max,LIM3}}$, the maximum floe size in the sea ice model is thus set equal to $D_{n^*}$, the upper bound of the $n^*$th category. To force the FSD to follow this power-law during the computation of the mechanical redistribution term $\Phi_f$, in LIM3 we introduce changes in the computation of $\beta$ and $Q(D)$. When using $N$ floe size categories, the redistribution equation (A1) becomes:

$$\Phi_{f,n} = -Q_n g_n + \sum_{m=1}^{N} \beta(m,n) Q_m g_m \ , m \in [0,N] \tag{A6}$$

Following Zhang et al. (2015), the redistribution factor $\beta(m,n)$ must respect Eq.A2. $\beta(m,n)$ should also allow for a switch from completely unbroken ice to a truncated power-law distribution with lower limit $D_0$ and upper limit $D_{\mathrm{max,LIM3}}$ if fragmentation occurs. Finally, $\beta(m,n)$ must ensure that floe size can only decrease during the fragmentation. To do so, $\beta(m,n)$ is defined as:

$$\begin{cases} \beta(m,n) = \dfrac{D_n^{2-\gamma} - D_{n-1}^{2-\gamma}}{\min(D_{n^*}, D_m)^{2-\gamma} - D_0^{2-\gamma}} \ \text{if } m \geq n \text{ and } n \leq n^* \\ \beta(m,n) = 0 \text{ otherwise} \end{cases} \tag{A7}$$

With this choice of $\beta(m,n)$, the FSD of each floe size category $n < n^*$ is equal to the distribution function derived from the power-law assumed in WW3 ($g_{n,\mathrm{P.L}}$), given by:

$$g_{n,\mathrm{P.L}} = c \frac{\int_{D_{n-1}}^{D_n} D^2 p(D) dD}{\int_{D_0}^{D_{\max}} D^2 p(D) dD} = c \frac{D_n^{2-\gamma} - D_{n-1}^{2-\gamma}}{D_{\max}^{2-\gamma} - D_0^{2-\gamma}}, \tag{A8}$$

$c$ being the sea ice concentration.

If sea ice in a given grid cell has already been broken, the FSD may have deviated from the truncated power-law distribution (due to advection or melting). If fragmentation occurs again at a later model time step, we force the FSD to be reset to the power-law assumed in WW3, by adjusting the fraction of each floe size category contributing to the redistribution through the value $Q_n$. This ensures that the FSD in LIM3 and WW3 are identical. After a fragmentation event, $D_{\mathrm{max,LIM3}}$ is the new maximum floe size in LIM3. The sea ice contained in floe size categories associated with floes larger than $D_{\mathrm{max,LIM3}}$ is therefore entirely redistributed into smaller floe size categories by setting:

$$Q_n|_{n>n^*} = 1. \tag{A9}$$

The smallest floe size category (i.e $D \in [D_0, D_1]$) does not contribute to the floe size redistribution, assuming that this category accounts for floes too small to be broken by waves (Toyota et al., 2011). It therefore forces $Q_1 = 0$. For a given floe size category $n$, we define $\Delta g_{th,n}$ as the difference between the actual and theoretical values of the FSD for this floe size category

$(\Delta g_{th,n} = g_n - g_{n,\mathrm{P.L}})$, and the theoretical value is given by the truncated power-law between $D_0$ and $D_{\mathrm{max,LIM3}}$. After the redistribution of floes between categories, $\Delta g_{th,n}$ needs to be zero, which is achieved through the adjustment of $Q_n$ in order to obtain $\Phi_{f,n} = \Delta g_{th,n}$. The following system thus needs to be solved:

$$
\begin{cases}
\Phi_{f,2} = (-1 + \beta_{2,2})Q_2 g_2 + \beta_{3,2}Q_3 g_3 + ... + \beta_{n^*,2}Q_{n^*}g_{n^*} + \displaystyle\sum_{n>n^*}^{N} \beta_{n \geq n^*,2}g_n \\[4mm]
\Phi_{f,3} = (-1 + \beta_{3,3})Q_3 g_3 + \beta_{4,3}Q_4 g_4 + ... + \beta_{n^*,n^*}Q_{n^*}g_{n^*} + \displaystyle\sum_{n>n^*}^{N} \beta_{n \geq n^*,3}g_n \\[4mm]
... \\[4mm]
\Phi_{f,n^*} = (-1 + \beta_{n^*,n^*})Q_{n^*}g_{n^*} + \displaystyle\sum_{n>n^*}^{N} \beta_{n \geq n^*,n^*}g_n,
\end{cases}
\tag{A10}
$$

This system consists in a triangular matrix in which all diagonal terms are non-zero. It is solved by doing:

$$
\begin{cases}
Q_{n^*} = \max\left(0, \dfrac{\Delta g_{th,n^*} - \sum_{n>n^*}^{N} \beta_{n \geq n^*,n^*}g_n}{g_{n^*}(\beta_{n^*,n^*} - 1)}\right) \\[5mm]
... \\[5mm]
Q_2 = \max\left(0, \dfrac{\Delta g_{th,2} - \sum_{n>2}^{N} Q_n \beta_{n,2}g_n}{g_2(\beta_{2,2} - 1)}\right)
\end{cases}
\tag{A11}
$$

The constraint $Q_n > 0$ ensures that the redistribution can only be done towards categories containing smaller floe sizes. This constraint thus implies that, in the case where $\Delta g_{th,n} > 0$, the FSD in LIM3 is reset to the truncated power-law only if there is enough sea ice in large floe categories to be redistributed into smaller floe categories. Besides, setting $Q_1 = 0$ means that the sea ice concentration associated with the smallest floe size category is never redistributed. In the absence of lateral growth, a succession of fragmentation events leads to an accumulation of floes in this category, deviating the FSD from the theoretical power-law for floe sizes between $D_0$ and $D_1$ (see Fig. 3).

*Competing interests.* The authors declare no competing interests.

*Acknowledgements.* G.B. and F.A. are supported by DGA, ANR grants ANR-14-CE01-0012 MIMOSA, ANR-10-LABX-19-01, EU-FP7 project SWARP under grant agreement 607476, ONR grant number N0001416WX01117. Part of this work has been carried out as part of the Copernicus Marine Environment Monitoring Service (CMEMS) ArcticMix and WIzARd projects. CMEMS is implemented by Mercator Ocean in the framework of a delegation agreement with the European Union. We thank Martin Vancoppenolle for his valuable help as well as Verena Haid and Xavier Couvelard for their significant assistance in setting up the coupled framework.

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

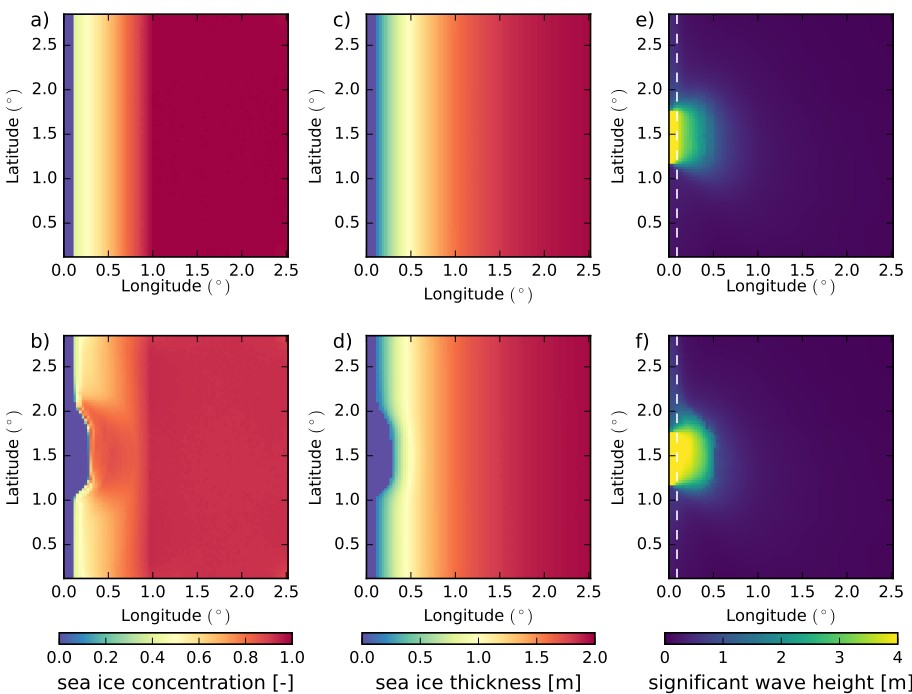

**Figure 1.** Implementation of the WRS in the idealized configuration. (a) and (c) show the initial state of sea ice concentration and thickness (ice-cover average), respectively. (b) and (d) show sea ice concentration and thickness after 72 h in the WW3-LIM3 coupled model. (e) and (f) show the significant wave height $Hs$ distribution after 72 h in the WW3 model and in the WW3-LIM3 coupled model, respectively. The white dashed line on (e) and (f) indicates the position of the ice edge ( $c$=0.15).

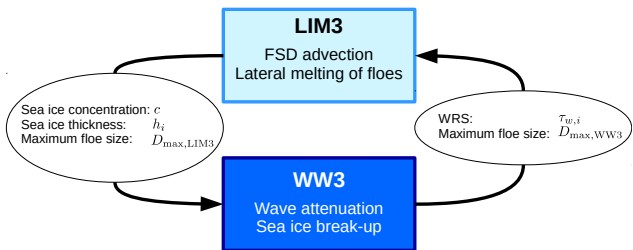

**Figure 2.** Schematic summary of the exchanged information between the sea ice model LIM3 and the wave model WAVEWATCH III® in our coupled framework. The two boxes correspond to the processes accounted for in a given model, while the variables exchanged between the models are listed in the bubbles.

.

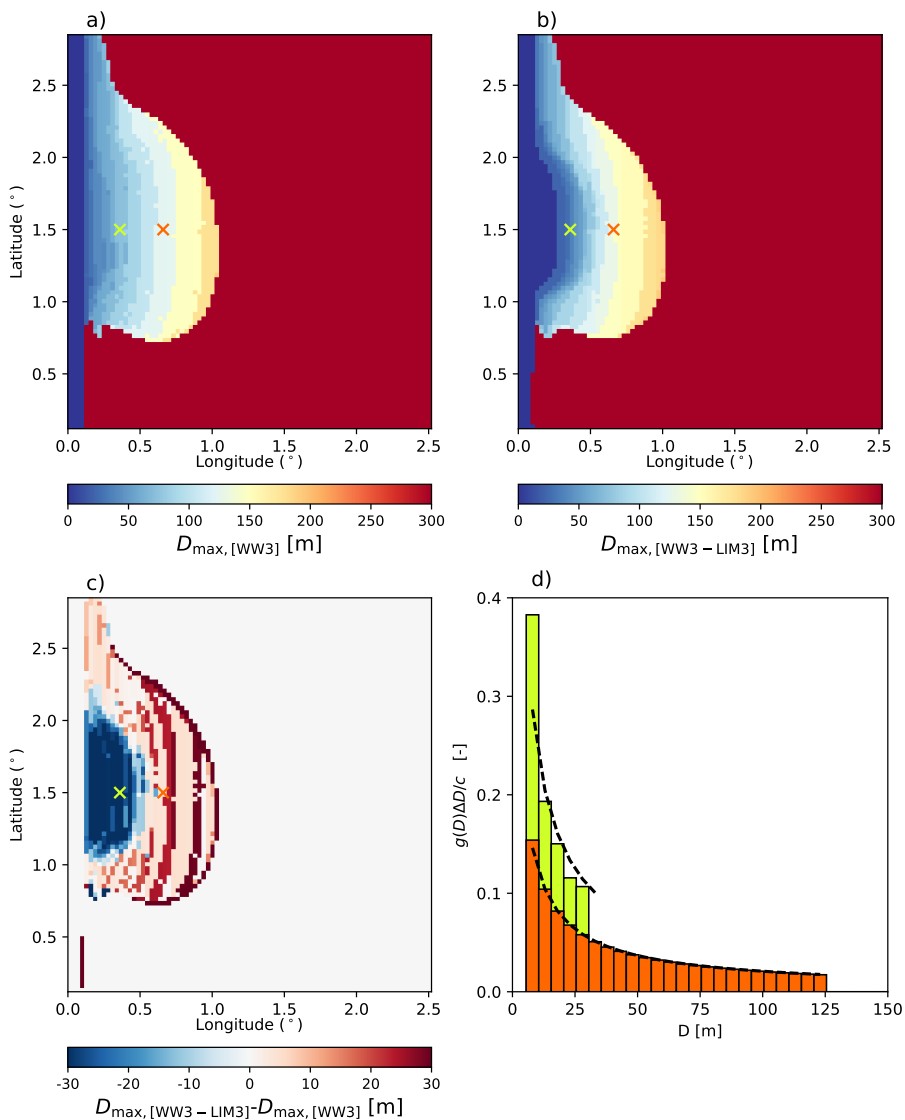

**Figure 3.** Snapshots of $D_{\mathrm{max}}$ from the uncoupled WW3 (a) and the WW3-LIM3 (b) simulations after 72 h, and the difference between the two (c). Panel (d) shows the FSD from the WW3-LIM3 run at two locations indicated with crosses on panels a,b,c. The black dashed line in (d) corresponds to the theoretical power-law FSD assumed in WW3.

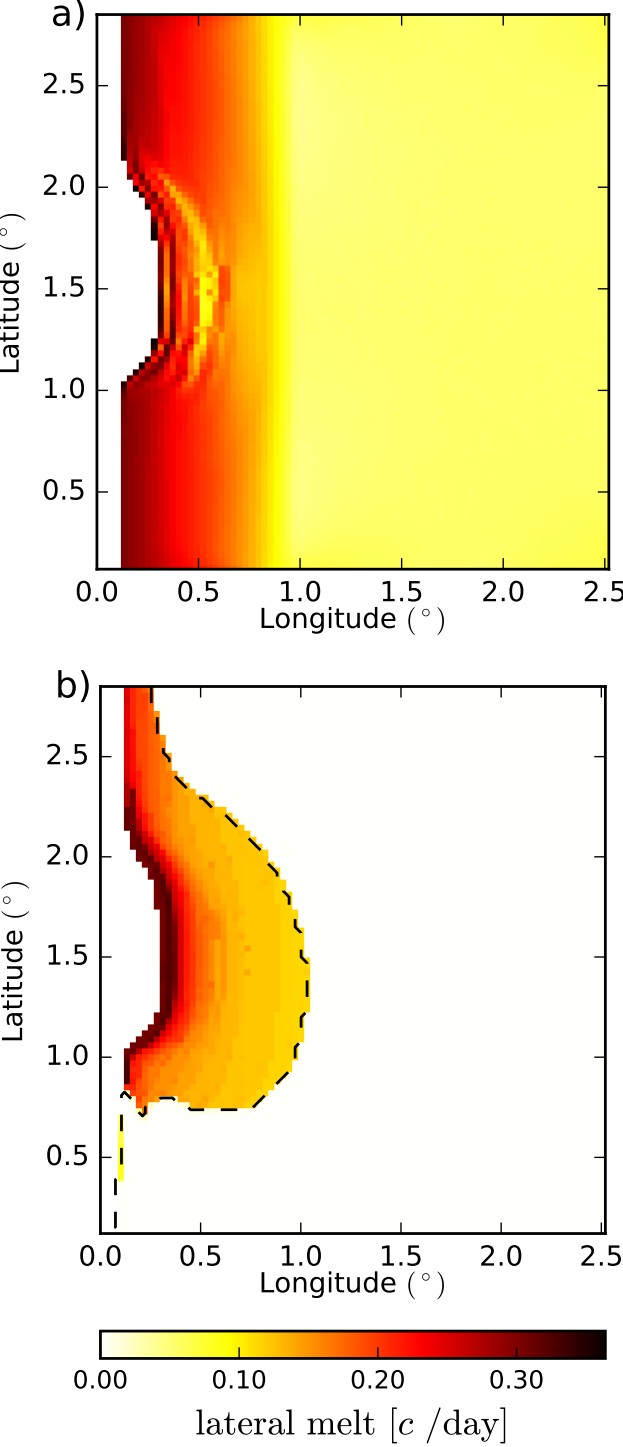

**Figure 4.** Lateral melt rates [estimated as percentage of sea ice concentration lost per day] estimated by the coupled model after 72 h of simulation using the parameterization of Lüpkes et al. (2012) (a), or the parameterization developed in this study accounting for wave-induced sea ice fragmentation (b). The black dashed contour on panel (b) indicates $D_{\mathrm{max}} = 500\,m$, and thus represents the limit between broken and unbroken sea ice.

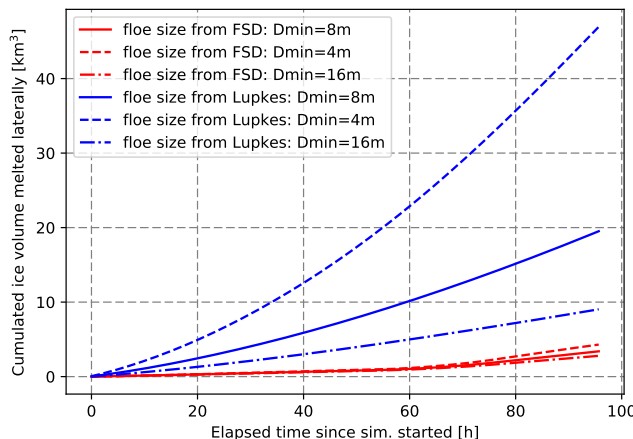

**Figure 5.** Temporal evolution of the sea ice volume loss due to lateral melt integrated over the whole domain for simulations similar to the one presented in Fig. 4, but for different values of $D_{\min}$. The two colors correspond to the two lateral melt parameterizations used in this study.

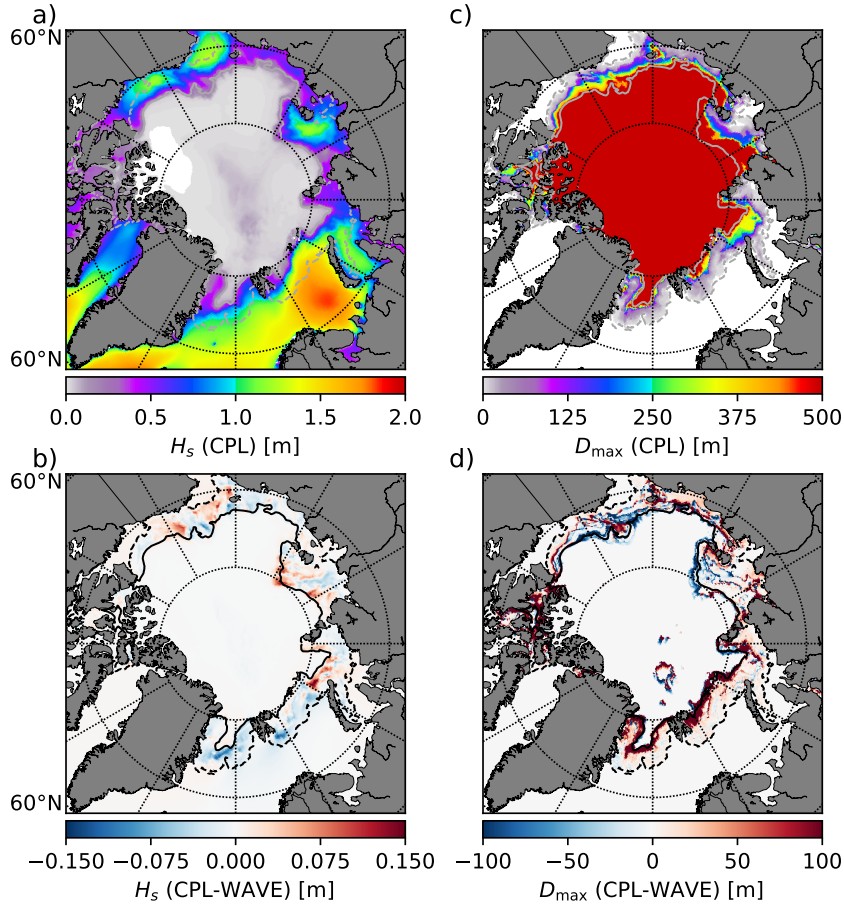

**Figure 6.** Significant wave height (a) and maximum floe size (c) in the CPL simulation averaged over the period 03-08-2010 to 09-09-2018, and the differences with the WAVE simulations (b, d). The black and grey contours delimit the MIZ in the CPL simulation, defined here as $0 < \langle D_{\max} \rangle < 700$ m. Note that the sea ice conditions from the NOT_CPL run are used as forcing for the WAVE run and are thus similar in the WAVE and the NOT_CPL runs.

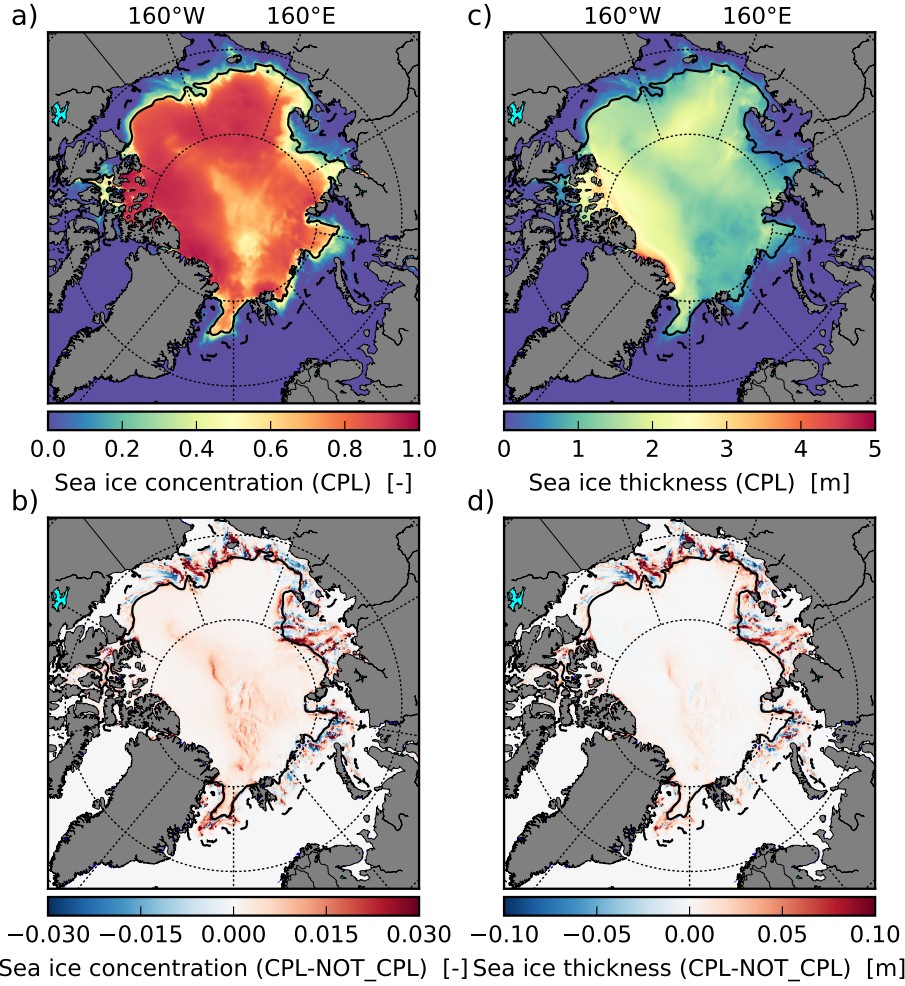

**Figure 7.** Sea ice concentration and thickness in the CPL simulation (a, c) and the difference with NOT_CPL (b,d) averaged over the period 04/08/2010 to 09/09/2010. The black contours delimit the MIZ in the CPL simulation, defined here as $0 < \langle D_{\max} \rangle < 700$ m.

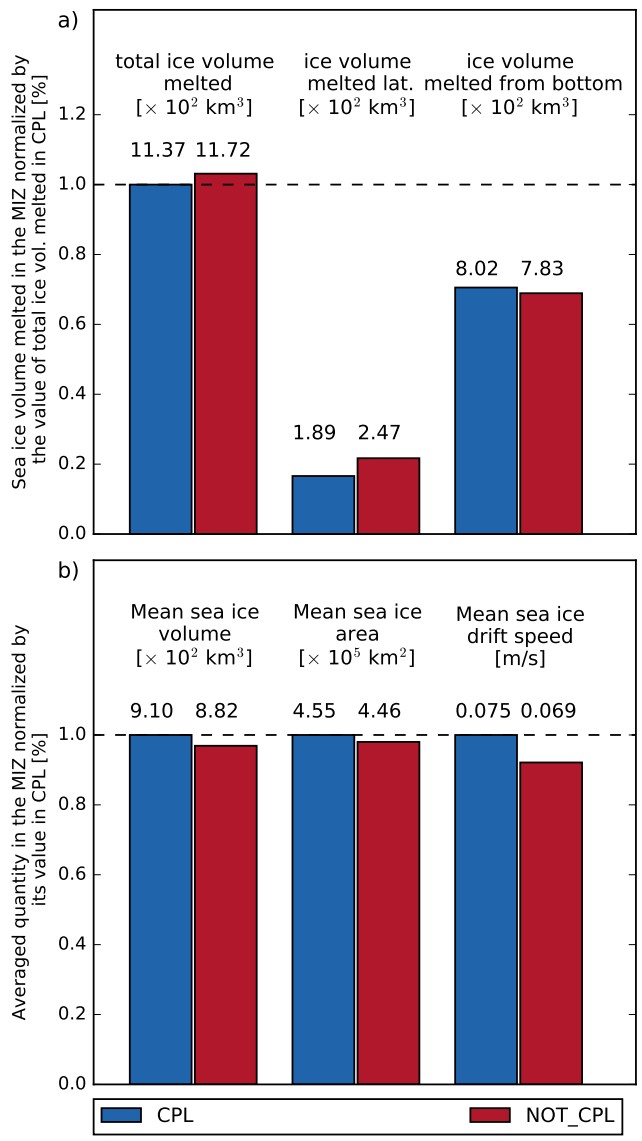

**Figure 8.** (a) Sea ice volume melted (in $10^2$km$^3$) integrated over the MIZ and over the period between 04/08/2010 and 09/09/2010 in the CPL and the NOT_CPL simulations. Here the MIZ is defined as the region where $0 < \langle D_{\max} \rangle < 700$ m in the CPL run. The contribution from lateral melt and bottom melt to the total melt for both simulations are also represented. (b) Mean sea ice volume (in $10^2$km$^3$), area (in $10^5$km$^2$), and drift speed (in m/s) in the MIZ over the same period. Values for each simulation are found above their associated bar.

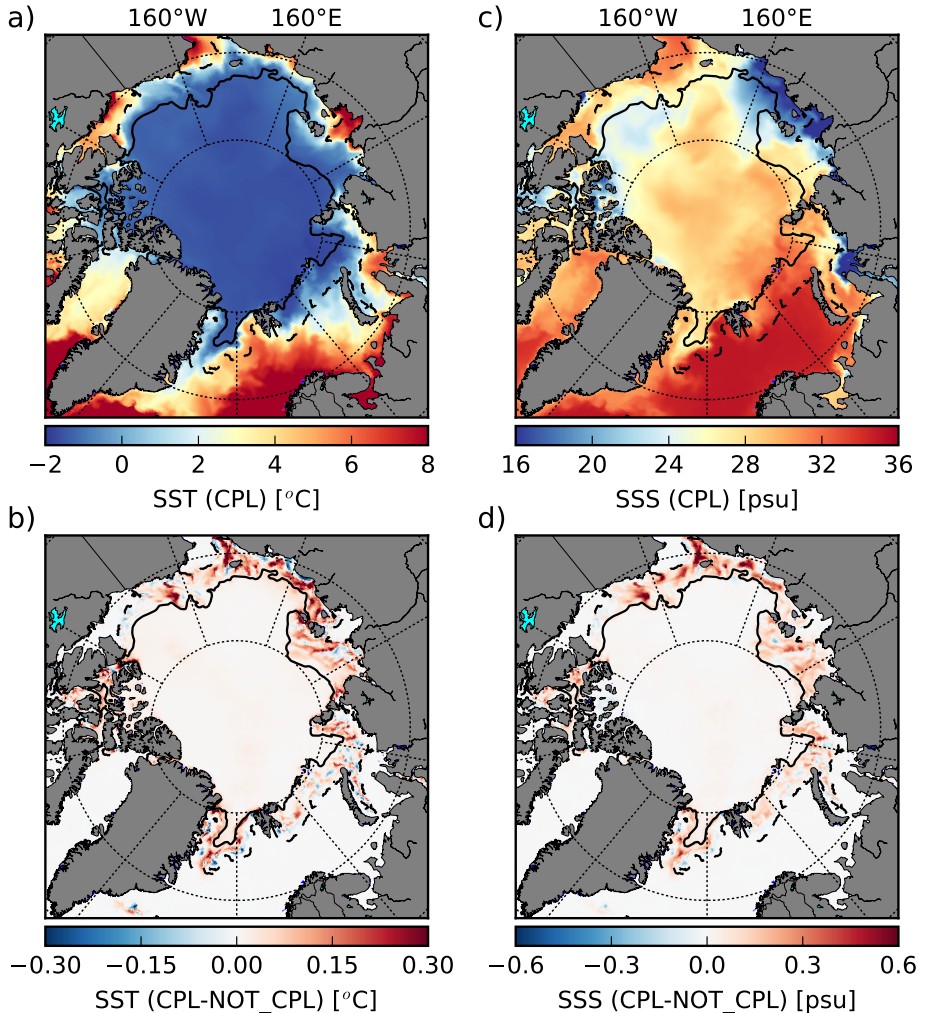

**Figure 9.** SST (a) and SSS (c) in the CPL run for the period between between 04/08/2010 and 09/09/2010, and the difference with NOT_CPL (b,d). The black contours delimit the MIZ in the CPL simulation, defined here as $0 < \langle D_{\max} \rangle < 700$ m.

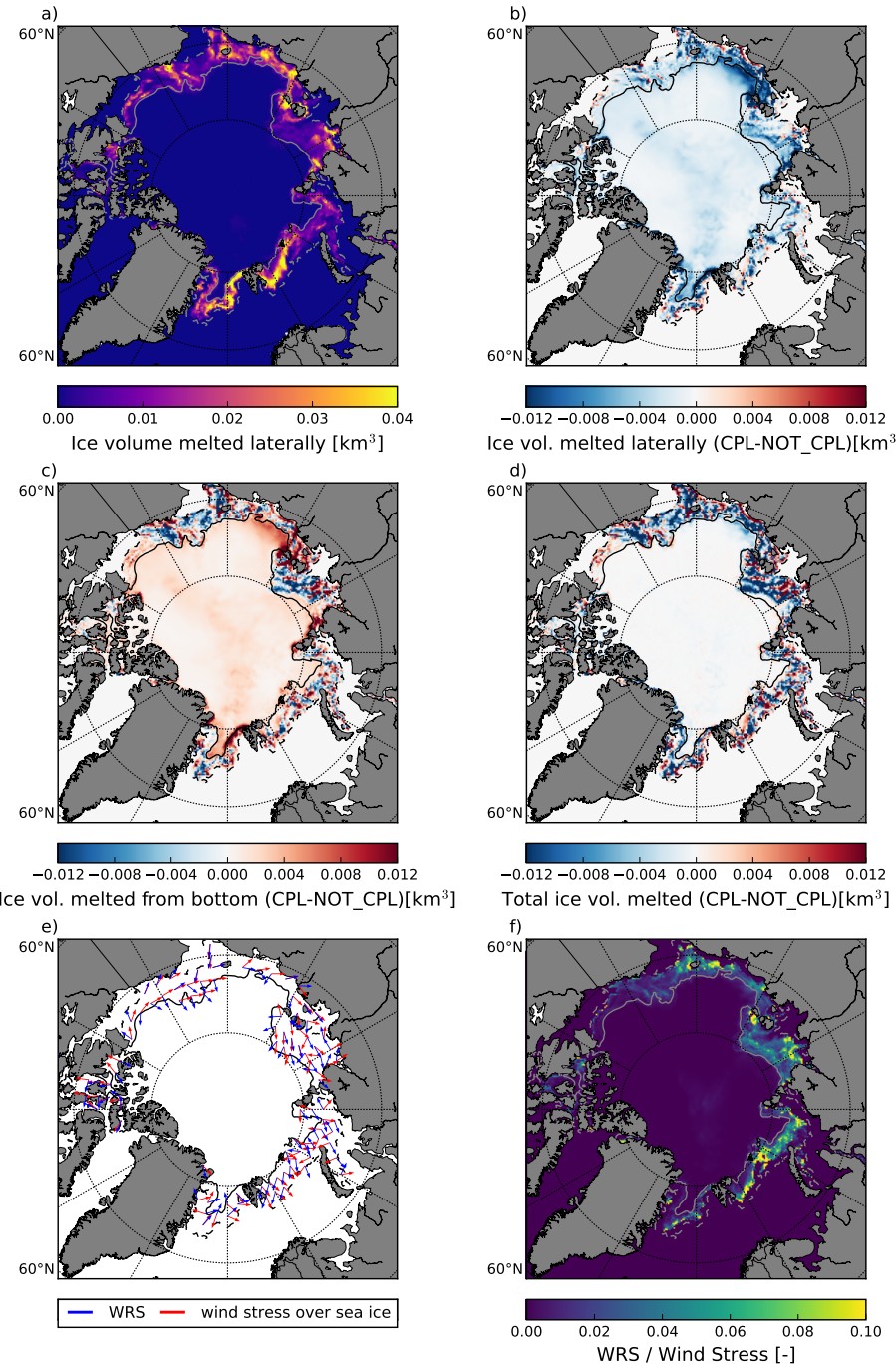

**Figure 10.** Volume of sea ice melted by lateral melt in the CPL simulation over the period between 04/08/2010 and 09/09/2010 (a). Differences between the CPL and the NOT_CPL runs of lateral melt (b), bottom melt (c) and total melt (d). (e) Wind stress (red) and WRS (blue) averaged over the same period 04/08/2010 and 09/09/2010 in the CPL simulation. Note that the WRS has been multiplied by a factor of 10 in order to improve readability. (f) Distribution of the relative magnitude of WRS over the wind stress. The grey contours represent the position of the ice edge ($c = 0.15$) on the first (solid line) and last day (dashed line) of the period considered in the CPL simulation.

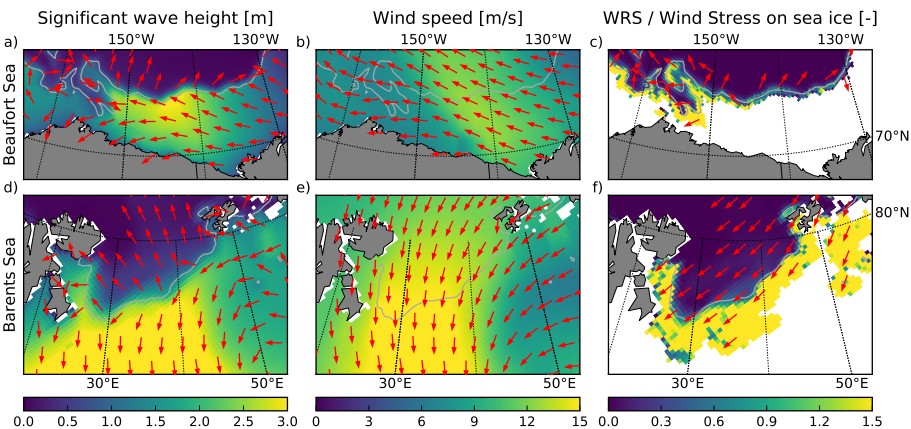

**Figure 11.** Significant wave height and wave mean direction of propagation (a, d), wind speed (b, e) and WRS (c, f) simulated by the CPL run during the storms that occurred in the Beaufort Sea (a, b, c) and in the Barents Sea (d, e, f) on 16/08/2010-17/08/2010. The grey contours indicate the position of the sea ice edge determined from the averaged sea ice concentration ($c = 0.15$).

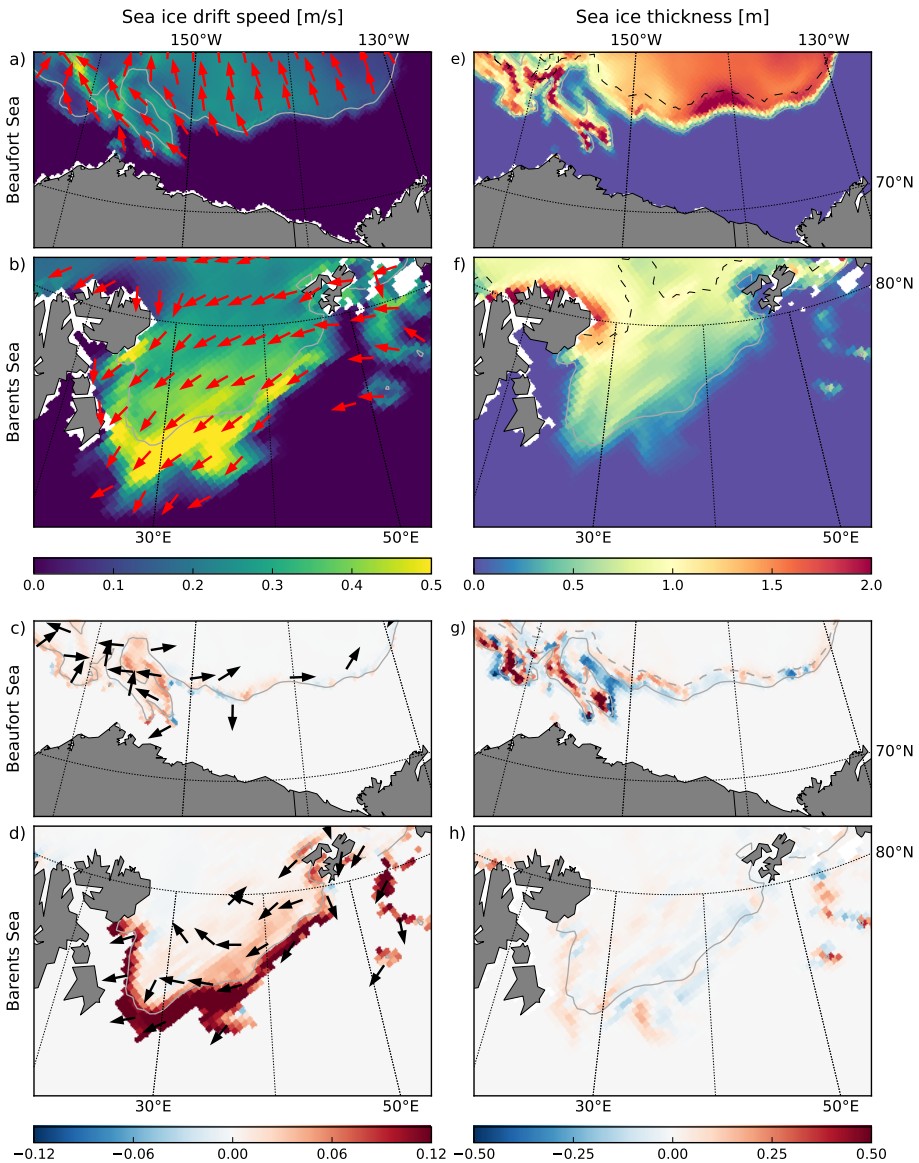

**Figure 12.** Mean sea ice drift (a, b) and sea ice thickness (e, f) simulated by the CPL run during the storms that occurred in the Beaufort Sea (a, c) and in the Barents Sea (b, f) on 16/08/2010-17/08/2010. Panels (c, d, g, h) show the differences for these quantities between the CPL and NOT_CPL simulations. Grey contours indicate the position of the ice edge determined from the averaged sea ice concentration ($c = 0.15$). The black dashed contour delimits the border between broken and unbroken ice ($D_{\max} = 500$m)

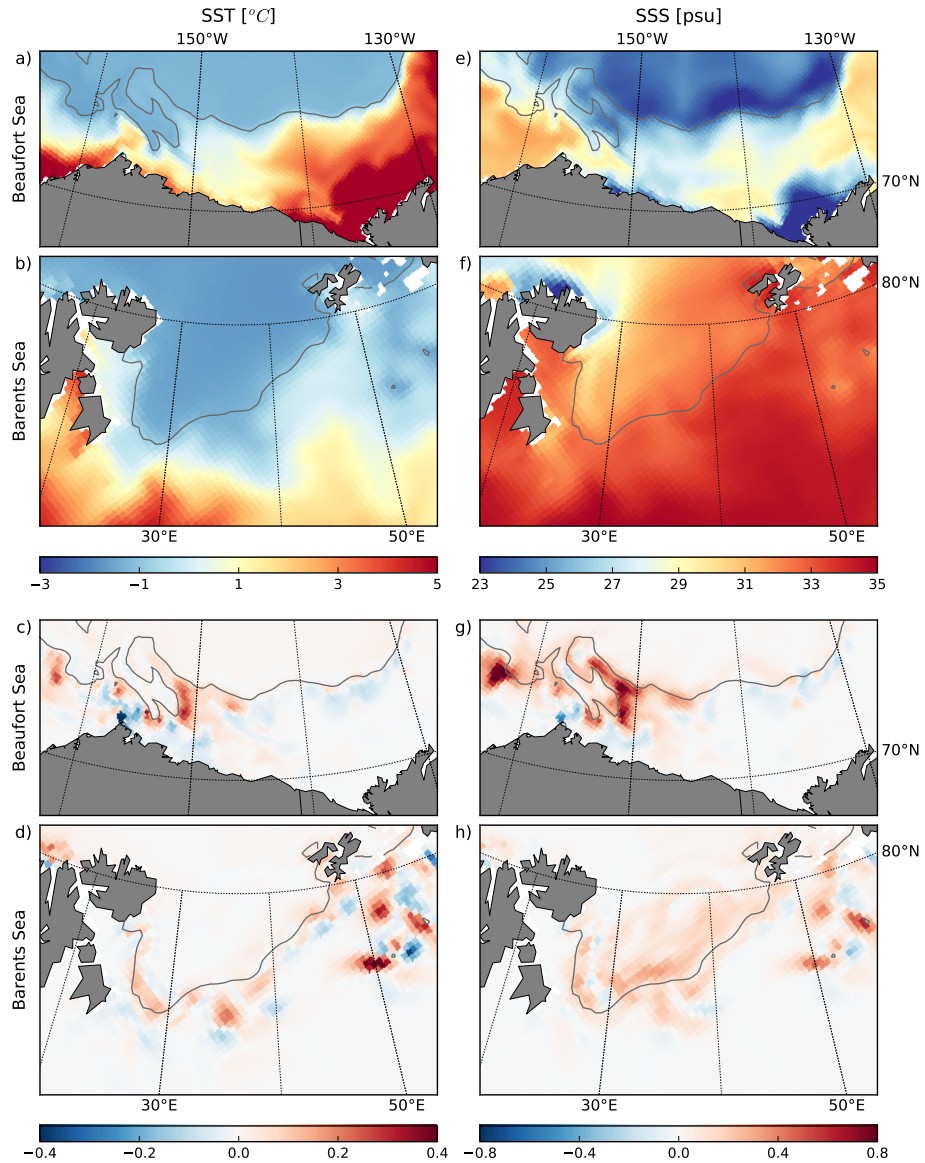

**Figure 13.** SST (a, b) and SSS (e, f) simulated by the CPL run during the storms that occurred in the Beaufort Sea (a, c) and in the Barents Sea (b, f) on 16/08/2010-17/08/2010. Panels (c, d, g, h) show the differences for these quantities between the CPL and NOT_CPL simulations. Grey contours indicate the position of the ice edge determined from the averaged sea ice concentration ($c = 0.15$).

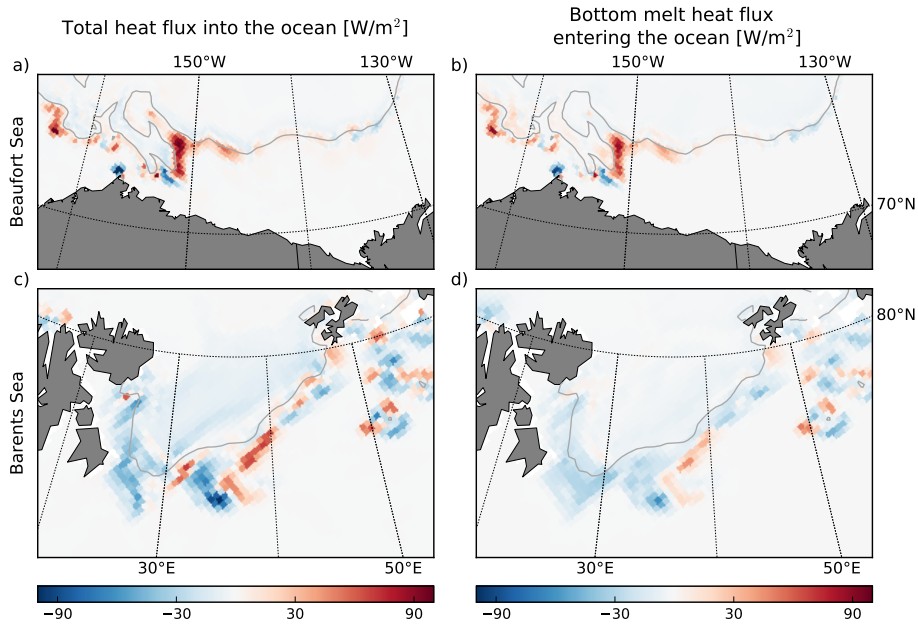

**Figure 14.** Averaged differences between the CPL and NOT_CPL simulations of (a, c) the heat flux into the ocean and (b, d) the contribution to the heat flux into the ocean coming from the sea ice bottom melt during the storms in the Beaufort Sea (a, b) and in the Barents Sea (c, d) which occurred on 16/08/2010-17/08/2010. Grey contours indicate the position of the ice edge determined from the averaged sea ice concentration ($c = 0.15$).