# Peer review of "Towards a coupled model to investigate wave-sea ice interactions in the Arctic marginal ice zone"

_The Cryosphere, 2019_

## Referee Comment (RC1) · Anonymous Referee #1 · 30 Jun 2019

This is an excellent paper. It clearly demonstrates the need to include ocean wind wave effects as part of a sea ice modelling frame work. The paper does not attempt to include all possible wave effects, for all situations, but rather, it limits itself to the role of the momentum flux due to waves attenuation by sea ice and the role of wave-induced sea ice break-up in lateral melt. There are many more steps towards the full inclusion of ocean waves into the sea ice modelling framework but this is a good first step.

Minor corrections:

Page 1, line 22: consider adding Waseda et al. 2018 to Thomson and Rogers 2014 Waseda et al. (2018): Correlated Increase of High Ocean Waves and Winds in the

[Figure]

Ice-Free Waters of the Arctic Ocean. Scientific Reports, 8, Article number: 4489 https://www.nature.com/articles/s41598-018-22500-9 Page 2, line 4: consider adding Bateson et al. (2019) Adam W. Bateson et al, 2019: Impact of floe size distribution on seasonal fragmentation and melt of Arctic sea ice. https://www.the-cryosphere-discuss.net/tc-2019-44/

Page 4, line 27: some fetch for the generation of sea ice. It is what you mean or rather some fetch for the generation of sea waves (?) Page 5, (1). I assume that Sice is defined as being positive, hence in WW3, it appears as –Sice. Just clarify. Page 6, line 9: panels b,e → panels b,d Page 10, line 19 : there is no figure 5 e nor (line 21), 5 f

---

## Referee Comment (RC2) · Anonymous Referee #2 · 1 Jul 2019

The manuscript describes a coupling between the Wavewatch III ocean surface wave model and the LIM3 sea ice model, which accounts for (i) wave radiation stress (WRS) on sea ice and (ii) floe-size-dependent lateral melt. Results are presented from an idealized model configuration, demonstrating the impact of individual steps in model development, and then from a pan-Arctic configuration where the sea ice model is coupled to NEMO. The authors find that (i) has an impact on sea ice dynamics on the order of 10%, and (ii) results in generally lower lateral melt than a parametrization based on sea ice concentration. They include interesting regional case studies as well as a pan-Arctic evaluation. The case studies highlight the importance of the direction of the WRS in determining its impact on the sea ice edge.

[Figure]

Coupling a sea ice and an ocean surface wave model is a valuable step forward, enabling investigation of marginal ice zone physics as well as potential advances for both sea ice and wave forecasting. The results on the impact of wave radiation stress on sea ice are very interesting, and certainly worth publication. In fact, the novelty of including this process in a commonly-used, pan-Arctic sea ice model should be emphasized further in the text. However, the manuscript includes some unclear reasoning and the floe size distribution model developed to examine the impact of lateral melt raises some questions that I list below in 'Specific Comments'. I also noted some incorrect representation of the literature.

In general, the manuscript is hard to follow, uses inaccurate or informal phrasing in places and contains a number of grammatical and typographical errors. It requires a thorough proof-read before re-submission. I have listed some sentences to be rephrased at the end of the review, but note that this is not an exhaustive list. When re-writing, the authors should carefully check where the text can be made clearer and more concise.

Specific Comments

P1 L2 and P1 L20: Strong & Rigor (2013) find that the Arctic MIZ (defined by sea ice concentration) has been expanding in summer and contracting in winter over the recent historical period. This should be referenced in the text.

P1 L2: 'Yet, state-of-the-art models are not capturing the complexity of the varied processes occurring in the MIZ, and in particular the processes involved in the ocean-sea ice interactions.' This is a very broad and vague sentence. The models may not include certain processes that occur in the MIZ, but they may be able to capture their large-scale impacts through parametrizations.

P1 L15-19: I would suggest the authors be more specific here about what processes they are referring to.

P2 L20: Check the location of reference placement in these sentences.

P2 L31: I would dispute the phrasing that 'In contrast, little progress has been done regarding the inclusion of waves in coupled ocean-sea ice models.' Simulation of the FSD within a climate-scale sea ice model is the first step required to model fracture of sea ice by ocean surface waves, and the past few years have seen much progress in this area: Zhang et al. (2015,2016), Horvat & Tziperman (2015), Bennetts et al. (2017), Roach et al. (2018), Bateson et al. (2019, in review). These studies have used simple representations of waves in order to develop the physics relating to sea ice, just as the studies focusing on the impact of sea ice on waves (Dumont et al. 2011, Williams et al. 2013, etc) have prescribed sea ice conditions and/or neglected certain sea ice physics. These paragraphs should be rewritten to more accurately reflect the current state of the literature, including all references I listed above. Additionally, I would use 'simple' rather than 'crude', which has negative connotations.

P3 L16: It would be useful to include a brief summary of the different processes by which sea ice affects waves in the model, for readers not familiar with the Boutin et al. (2018) paper.

P4 L1: What does 'Arctic realistic simulation' mean?

P4 L24: Describe the Lupkes et al. (2012) parametrization at its first mention, or don't mention it here.

P4 L23: If I understand correctly, the full NEMO ocean model is initialized from a climatology and spun-up for nine years. This seems to be a rather short spin-up period. How was it determined that nine years was sufficient? Similarly, how was it determined that three days was a sufficient adjustment period for the introduction of the wave coupling?

P5 L12: 'Updated floe size.' how is 'floe size' defined?

P5 L14: 'floe size is actualized'. What does this mean? Also, P9 L11: What is 'actual

floe size'? Similarly P9 L25: What is the 'actual FSD'?

P5 L14: 'LIM3 takes into account the WRS in its ice transport equation'. This should be stated in the Introduction, as it is a key contribution of the manuscript.

P6 L4: How is the partial sea ice cover already accounted for in WW3?

P6 L22: Define the sea ice thickness distribution and the FSD function. Is the latter an areal distribution?

P6 L22: I think a little more explanation would be useful here for readers not familiar with the various FSD schemes in the literature. Add a sentence or so on why the Zhang et al. (2015) approach is chosen over the Horvat & Tziperman (2015) approach. The sentence from P18 L13 'assuming floes of different sizes...' should be stated here as well.

P6 L28: 'implemented a FSD that enables floes to be advected..' – the FSD itself does not enable this, presumably this should say that Williams et al. (2017) implemented a scheme for advection of the FSD. Consider summarizing how this works e.g. what quantity is advected?

P6 L31: 'We do not make any assumption on its shape in general, but the FSD is forced to follow the power-law assumed in WW3 as soon as wave-induced sea ice break-up occurs.' This sentence seems somewhat self-contradictory: there is an assumption on its shape if the FSD is constrained to follow a power-law.

P7 L4: 'Assuming a power-law FSD is coherent with a distribution caused by a succession of break-up events (Toyota et al., 2011, Dumont et al. 2011).' The Toyota et al. 2011 study finds a change in the value of the exponent of a power-law fit to their data at around 40m. Does the model presented here assume a single power-law exponent, or include this transition? Also note that the Toyota et al. 2011 study covers a small area in space and time, and therefore may not be globally applicable. The Dumont et al. 2011 study itself does not show that a power-law FSD arises from a succession of

break-up events, but rather provides a mathematical description for this assumption, so this citation should be removed or discussed in a different way.

P7 L8-22: The authors assume a power-law FSD in WW3 and then force LIM3 to follow the same power-law when wave fracture occurs. As they state, the effects of sea ice advection and thermodynamics cause deviations from a power-law. However, the effects of these processes may be over-ruled to continue to force the FSD to follow a power law, at a frequency determined by an arbitrary parameter. I don't understand why the authors take this approach. Why include other FSD processes if they are not always allowed to affect the FSD? How often does such over-ruling occur - is this most of the time or in a small fraction of timesteps? Is there an alternative approach to the power-law assumption? The assumption has not been well justified in the manuscript.

Similarly, can sea ice fracture be handled in the sea ice model rather than the wave model? I would have thought that this would avoid the need for the Dmax adjustment.

P8 L3: 'This sensitivity remains really small.' This statement should be quantified more precisely, and the authors should describe how they determined this or consider adding their sensitivity results to Supplementary Material. Was sensitivity to the smallest resolved floe size tested? I would expect that lateral melt would be particularly sensitive to this.

P8 L16: Is this the same experiment as in Fig. 1? It would help the reader to restate what the differences in the two runs correspond to physically. I think there are quite a few differences - evolving sea ice, advection of Dmax - which make it hard to understand what the differences in the model output mean.

P9 L8: The Lupkes et al parametrization should be defined explicitly. Is this what LIM3 uses as standard for D in Eqn. 5? Please explain the reason for using it here.

P9 L11: A situation where ice concentration is less than 0.6 and floe size is greater than 10 m could occur anywhere, for example near the ice edge in wave-free conditions, so

I suggest removing the first part of this sentence.

P9 L30: As stated above, I would expect the amount of lateral melt to depend strongly on Dmin. Have the authors investigated this? If not, the results on lateral melt should include some discussion of this.

Section 4.1: This section compares the CPL and WAVE simulations at the pan-Arctic scale. The differences between the simulations include the impact of wave radiation stress and floe-size-dependent lateral melt. The authors then try to attribute various impacts to one of these two processes. Why not consider two separate runs here, one which adds the wave radiation stress only and one which adds the floe-size-dependent lateral melt only? As it is, I found it difficult to understand this evaluation.

Section 4.1 in general was difficult to read. The text usually described differences to the CPL run. However, the WAVE and NO-CPL runs should be considered as the reference simulations, and so differences should be described in the CPL run relative to the reference runs (i.e. describe an increase in CPL relative to NO-CPL, rather than a decrease in NO-CPL). I think this would improve the readability.

P10 L19: It would help the reader to briefly restate the differences in the runs at the start of the paragraph, including the note at L27 ('One should keep in mind...). Also note mis-matched parentheses here.

Sec. 4.1.3: The discussion of lateral melt would be aided by figures showing some equivalent floe size statistic from the Lupkes parametrization and from the FSD model.

P12 L11: 'This result does not reflect the fact...' What does this sentence mean?

P12 L17: 'Actually, in contrast to what was found in previous studies by Zhang et al. (2016), Bennetts et al. (2017), Roach et al. (2018a), de-activating completely lateral melt in both runs (not shown) has a negligible effect on the quantity of melted ice in our simulations (not shown).' The three named studies did not deactivate lateral melt, so the results presented here cannot be 'in contrast' to theirs. However, Roach, Dean

and Renwick (2018) did essentially deactivate lateral melt, by setting all floe sizes to 10000m, and showed that this had no impact on sea ice concentration in the Antarctic.

Section 4.2: This subsection is very interesting, but again hard to follow. Perhaps consider using one figure for each case, reducing the number of variables shown in figures in the main body of the paper, and moving the remainder to Supplementary Information. More figures could be added in the Supplementary for some of the 'not shown' aspects. I counted thirteen 'not shown' aspects in the paper, which seems rather high.

P16 L13: 'It is, however, mostly compensated by an increase of lateral melt.' Add that this is the converse of what has been shown in previous studies.

P16 L8: 'The coupled model was then used to examine . . .. the effects of wave-induced sea ice break-up on sea ice melt.' Rather, the study compares their model to an alternative parametrization for lateral melt (the Lupkes parametrization), that is designed to approximate varying floe sizes for different concentrations. To isolate the impact of the wave-induced break-up, or the 'impact of the coupling' as mentioned earlier, a more suitable comparison would be to a simulation where all floes were unbroken. Otherwise, modify the discussion in the text.

P16 L30: Similarly, the paragraph at P16 L30 compares the difference in lateral melt between the FSD model and the Lupkes parametrization (with varying floe size) to the differences found in previous studies. However, these previous studies show differences between a FSD model and a constant floe size parametrization for lateral melt, so should not be directly compared to this study. The discussion of the various studies should reflect this.

P17 L7: 'One should also remember that the studies of Zhang et al. (2016) and Roach et al. (2018b) were aiming at representing the evolution of floes larger than 1000 m.' This is incorrect. Both studies represent floes up to a maximum floe size of around 1000 m (radius). Also note that Roach et al. (2018a) and Roach et al. (2018b) are

confused in places.

P17 L13: 'Among the wave-sea ice interaction processes...' This sentence is unclear. Impact on what?

Presentational Comments

Throughout, I would suggest referring to 'ocean surface waves' in the abstract and early parts of the Introduction, rather than simply 'waves' for clarity.

I would also suggest using 'sea ice fracture' rather than 'sea ice break-up,' as this is used in other studies

In general, the definite article is over-used e.g. 'the sea ice near the sea ice edge' can simply be 'sea ice near the sea ice edge', 'impact the sea ice floe size' can be 'impact sea ice floe size' etc.

P1 L3: 'In the present study....' - clumsy sentence, suggest rewording

P1 'highlight the need to include the wave-sea ice processes in models aiming at forecasting sea ice condition on short time scale' -> 'highlight the need to include wave-sea ice processes in models used to forecast sea ice conditions on short time scales'

P2 L5: -> 'and sea ice drift'

P2 L12: 'in the direction of the propagation'

P2 L13: 'Southern ocean' -> 'Southern Ocean'

P2 L14: 'may become more prominent in the Arctic in the future.'

P2 L27: 'a first step was done' -> 'a first step was made', similarly elsewhere progress is 'made' rather than 'done'

P3: reword 'wave by sea ice'; 'is implemented or not'; 'without any wind or ocean current'; also the sentences on timesteps

[Figure]

P3 L14: change 'on' to 'of'

P4 L20: 'aim at compensating' -> 'was made to compensate'

P4 L31 'in this particular year'

P4 L31: reword 'storms occurring during it'

P5 L1: 'referred to as WAVE'

P5 L9: sentences about average thickness - seem to use a lot of words to say something fairly straightforward

P5 L29: define vector k

P7 L8: 'the coupling between the two models can be done' -> 'the two models can be coupled'

P10 L4: The introduction to Section 4 seems unnecessarily lengthy and should be made more concise.

P10 L5: 'the impact of the including the wave-sea ice interactions' - reword

P13 L25: 'that is exposed upwind (and waves)' - reword

P14 L19: 'could in principle modified' - reword

P14 L34 'very high waves of which attenuation induces WRS' - reword

P15 L14: 'pattern than' -> 'pattern to'

P15 L32 'low concentrated' -> 'of low concentration'

P16 L15: 'generating higher and more energetic waves'

P17 L7: 'were aiming at representing' -> 'aimed to represent'

P17 L14: 'additional lateral source melt' - reword

Section 4.1 figures – in the reference plots, I found the colormaps rather counter-intuitive. Consider choosing maps that are white at zero.

Fig. 7: y-axis label lists the units as %, but values on the y-axis are out of 1. I presume that the $10^2$ km$^3$ corresponds the numbers on the figure, but this should be noted in the legend.

---

## Referee Comment (RC3) · Anonymous Referee #3 · 12 Aug 2019

The manuscript Toward a coupled model to investigate wave-sea ice interactions in the Arctic marginal ice zone by Boutin et al. presents a model that couples waves and sea ice dynamics to study the impact of waves on sea ice evolution over the Arctic Ocean. The model includes a floe size and thickness distribution as a prognostic variable that is exchanged between the sea ice and wave components. The FSTD obeys an evolution equation that includes floe-size dependent processes such as lateral melt and wave break-up. A focus is put on the wave radiation stress arising from wave attenuation in sea ice that imposes an additional force on the ice, and on the floe-size dependent lateral melt parameterization. The impact of wave-related processes on sea ice are studied by comparing simulations of NEMO-LIM3 (ice-ocean component) that

is coupled and uncoupled to WW3 (wave component) over a pan-Arctic domain, and during two storm case. The comparison is done over a month-long period, at the end of summer 2010, after a 8-year spin-up period.

Overall the paper makes a significant contribution to the modeling of polar marine environment in the sense that it provides a very useful tool to study the complexities wave-ice interactions and their impact over different spatio-temporal scales. The discussion puts the study in the context of the recent developments and describes the limitations, thus pointing towards important issues to be addressed in order to make further progress (duration of the simulation, atmospheric and oceanic coupling, floe-size dependent ice rheology missing, freezing period not studied, etc.). It is well written, despite some typos and corrections that need to be made, and descriptions of model implementation and results are detailed enough, although some key information is missing (see below). It is thus worthy of publication, after minor revisions are made.

Specific comments

P4. L18. Wave attenuation is a central piece of the study, as it determines the wave radiation stress and, to a certain extent, the extent of the wave-induced ice break-up area (i.e. the marginal ice zone). Because of this, I suggest that in addition to referring to Ardhuin et al. (2018) for the choice of the wave attenuation, authors recall the main characteristics of the attenuation scheme. Is it floe-size and/or thickness dependent, and how? Is it a dissipative or scattering scheme (or a mix of both)? This could be done in a few lines.

P6. L1. Another central piece of the study is the ice drift resulting from the momentum balance. Here the WRS is added as an external forcing term that will be balanced by the internal stress, and model solutions may depend strongly on rheology parameters. I understand that this term (rheology) has not been modified significantly from what's typically used by LIM3 users, and that studying the ie rheology is not the focus of the paper, but it needs to be described minimally here. The rheology contains a few

parameters that can be tuned for various reasons, including the compressive strength, the shear-to-compressive strength ratio, if not the yield curve itself or the numerical scheme. Describe what rheology is used and what are the main parameter values. Maybe adding a table would serve well that purpose.

P11. L14. Warmer and saltier surface waters in the CPL run seems to point towards that enhanced turbulent mixing arising by increased shear stress between the ice and the ocean, dominates over enhanced melting, which tends to produce fresh and cold anomalies. The following section focuses on an interpretation of that response in terms of the differences between the lateral melt parameterization. Have you looked at mixing as a possible mechanism for explaining it? Are there anomalies in the mixing or mixed layer depth in the marginal ice zone? This mechanism is discussed very clearly later in the two storm cases, but it would be interesting to discuss it also for the pan-Arctic case.

P19. EqA3. Define $D_*$. And later, define also $n_*$. Is $D_*$ equivalent to $D_{n*}$?

Some typos

P5. L14. Replace actualized by updated.

P5. L20. ... is transferred to what has caused this attenuation.

P5. Eq2. Remove parentheses around $\sigma$.

P6. L22. multi-category.

P7. L29. $c$ has already been introduced as the concentration earlier.

P8. L10. Is Toyota et al. (2011) the right reference for this statement? There are older and more appropriate references for this it seems. The smallest floe size that can be generated by flexural break-up is thickness-dependent. Maybe this should be acknowledged.

P8. L25. Uncoupled instead of not coupled (also at various other place in the

manuscript).

P9. L8. Based on a number of observations.

P9. L17. Rather than on sea ice conditions.

P10. L8. . . . on sea ice conditions.

P10. L19. There is no panel e on Fig. 5.

P11. L9. Do you refer to the grid cell average thickness? Specify.

P11. L11. There are also differences . . .

P12. L1. . . . property anomalies.

P14. L6. Difference (singular).

P17. L24. when trying to forecast . . .

Fig2. Schematic summary of . . . The two boxes correspond . . .

Fig3. Panel c. notcpl should be replaced by NOT_CPL in the index. You can also specify the run elsewhere than in the index to avoid expanding indices.

Fig5. The black and grey contours . . .

---

## Author Comment (AC1) · 28 Sep 2019

We thank the reviewer for their careful reading of our manuscript and for their comments and suggestions. We have tried to address their questions and concerns in our response. Our comments can be found in the attached .zip file, along with an updated version of the manuscript and a document highlighting the different changes between the two versions.

Please also note the supplement to this comment:
https://www.the-cryosphere-discuss.net/tc-2019-92/tc-2019-92-AC1-supplement.zip

---

## Author Response (AR1)

**Referee 1**
**This is an excellent paper. It clearly demonstrates the need to include ocean wind wave effects as part of a sea ice modelling framework. The paper does not attempt to include all possible wave effects, for all situations, but rather, it limits itself to the role of the momentum flux due to waves attenuation by sea ice and the role of wave-induced sea ice break-up in lateral melt. There are many more steps towards the full inclusion of ocean waves into the sea ice modelling framework but this is a good first step.**

We thank the reviewer for their careful reading of our manuscript and for their comments and suggestions. We have tried to address their questions and concerns, as detailed in the following. In our comments, PXLY refers to page X line Y of the attached updated manuscript.

**Minor corrections:**

**Page 1, line 22: consider adding Waseda et al. 2018 to Thomson and Rogers 2014**
**Waseda et al. (2018): Correlated Increase of High Ocean Waves and Winds in theIce-Free Waters of the Arctic Ocean. Scientific Reports, 8, Article number: 4489https://www.nature.com/articles/s41598-018-22500-9**
We have added these references (P2L3).

**Page 2, line 4: consider adding Bateson et al. (2019) Adam W. Bateson et al, 2019: Impact of floe size distribution on seasonal fragmentation and melt of Arctic sea ice. https://www.the-cryosphere-discuss.net/tc-2019-44**
We have added the reference later in the text, but did so in some other places where it seemed more relevant to us (P3L8, P14L13...).

**Page 4, line 27: some fetch for the generation of sea ice. It is what you mean or rather some fetch for the generation of sea waves (?)**
It was indeed a typo and we have fixed it.

**Page 5, (1). I assume that Sice is defined as being positive, hence in WW3, it appears as –Sice.  Just clarify.**
Our definition of Sice has been clarified in the updated manuscript (P6L14).

**Page 6, line9: panels b,e→panels b,d**
Fixed
**Page 10, line 19 : there is no figure 5 e nor (line 21), 5 f**
Fixed

**Referee 2**

**Coupling a sea ice and an ocean surface wave model is a valuable step forward, enabling investigation of marginal ice zone physics as well as potential advances for both sea ice and wave forecasting. The results on the impact of wave radiation stress on sea ice are very interesting, and certainly worth publication. In fact, the novelty of including this process in a commonly-used, pan-Arctic sea ice model should be emphasized further in the text. However, the manuscript includes some unclear reasoning and the floe size distribution model developed to examine the impact of lateral melt raises some questions that I list below in 'Specific Comments'. I also noted some incorrect representation of the literature. In general, the manuscript is hard to follow, uses inaccurate or informal phrasing in places and contains a number of grammatical and typographical errors. It requires a thorough proof-read before resubmission. I have listed some sentences to be rephrased at the end of the review, but note that this is not an exhaustive list. When re-writing, the authors should carefully check where the text can be made clearer and more concise.**

We thank the reviewer for their careful reading of our manuscript and for their comments and suggestions. We have tried our best to address their questions and concerns, as detailed in the following. A careful proof-reading of the text has been done to improve the readability of the text. In our comments, PXLY refers to page X line Y of the attached updated manuscript.

**Specific Comments:**

**P1 L2 and P1 L20: Strong et Rigor (2013) find that the Arctic MIZ (defined by sea ice concentration) has been expanding in summer and contracting in winter over the recent historical period. This should be referenced in the text.**
We now refer to Strong et Rigor (2013) in the introduction (P2L2)

**P1 L2: 'Yet, state-of-the-art models are not capturing the complexity of the varied processes occurring in the MIZ, and in particular the processes involved in the ocean-sea ice interactions.' This is a very broad and vague sentence. The models may not include certain processes that occur in the MIZ, but they may be able to capture their large-scale impacts through parametrizations.**
The sentence has been changed to:
P1L2: "*Yet, state-of-the-art models exhibit significant biases in their representation of the complex ocean-sea ice interactions taking place in the MIZ.*"

**P1 L15-19: I would suggest the authors be more specific here about what processes they are referring to**
We have added a few examples:
P1L5*: "Indeed, the MIZ is characterized by a wide variety of processes resulting from the highly non-linear interactions between the atmosphere, ocean and sea ice: sea ice floe fragmentation and welding, lead opening and associated heat transfers, mesoscale and submesoscale features arising from strong temperature and salinity gradients (see Lee et al., 2012, for a review and references therein)...*"

**P2 L20: Check the location of reference placement in these sentences.**
This has been fixed.

**P2 L31: I would dispute the phrasing that 'In contrast, little progress has been done regarding the inclusion of waves in coupled ocean-sea ice models.' Simulation of the FSD within a climate-scale sea ice model is the first step required to model fracture of sea ice by ocean surface waves, and the past few years have seen much progress in this area: Zhang et al. (2015,2016), Horvat et Tziperman (2015), Bennetts et al. (2017), Roach et al. (2018), Bateson et al. (2019, in review). These studies have used simple representations of waves in order to develop the physics relating to sea ice, just as the studies focusing on the impact of sea ice on waves (Dumont et al. 2011, Williams et al. 2013, etc) have prescribed sea ice conditions and/or neglected certain sea ice physics. These paragraphs should be rewritten to more accurately reflect the current state of the literature, including all references I listed above. Additionally, I would use 'simple' rather than 'crude', which has negative connotations.**

We agree with the reviewer that this paragraph was too negative considering the amount of work recently done on FSDs in sea ice models. We therefore rephrased it to emphasize the step-by-step progress that have allowed us to perform this study. We also added missing references to the work of Roach et al. (2018) and Bateson et al. (2019).

P3L1: *"In parallel, progress has also been made regarding the inclusion of the effects of waves in coupled ocean-sea ice models. Using a very simple parameterization, Steele et al. (1989) and Perrie and Hu (1997) have investigated the effect of WRS on sea ice drift in the MIZ, only considering the attenuation of waves generated between the ice floes, and found a limited impact on the sea ice conditions. More recently, Williams et al. (2017) implemented a wave module in the semi-Lagrangian sea ice model neXtSIM (Rampal et al., 2016) and found that high wave conditions can cause a significant displacement of the sea ice edge. The implementation of FSDs in different sea ice models, as introduced by Zhang et al. (2015) and Horvat and Tziperman (2015) for instance, has also opened the way to the assessment of the potential enhancement of lateral melt by wave-induced ice fragmentation (Zhang et al., 2016; Bennetts et al., 2017; Roach et al., 2018; Bateson et al., 2019), but the representation of waves remains too simple to simulate the full effect of waves on the evolution of sea ice."*

**P3 L16: It would be useful to include a brief summary of the different processes by which sea ice affects waves in the model, for readers not familiar with the Boutin et al. (2018) paper.**

We have added the following sentence in section 2 to briefly describe the model detailed in Boutin et al. (2018):

P3L27: *"These processes are scattering (which redistributes the wave energy without dissipation), friction under sea ice (with a viscous and a turbulent part depending on the wave Reynolds number), and inelastic flexion. All these processes depend on sea ice thickness and concentration, and scattering and inelastic flexion also depend on floe size."*

**P4 L1: What does 'Arctic realistic simulation' mean?**

This sentence has been fully rephrased:

P4L17: *"The wave spectrum used as forcing at the boundary is extracted at a point south of Svalbard from an Arctic hindcast performed with WW3 described by Stopa et al. (2016). It covers the period of May 2nd to 3rd, 2010, during which a storm occurred in this particular area (Collins et al. 2015)."*

**P4 L24: Describe the Lupkes et al. (2012) parametrization at its first mention, or don't mention it here.**
We have removed this reference and modified the sentence as follows:
*P5L7:"[...], in which the already existing lateral melt parameterization in LIM3 is activated."*

**P4 L23: If I understand correctly, the full NEMO ocean model is initialized from a climatology and spun-up for nine years. This seems to be a rather short spin-up period. How was it determined that nine years was sufficient? Similarly, how was it determined that three days was a sufficient adjustment period for the introduction of the wave coupling?**
Regarding the ocean-sea ice model, the spin up would indeed be too short to allow for a full adjustment of the full water column. However, here, we only focus on ocean surface processes, that are expected to quickly respond to the atmospheric and sea ice forcing, for which 9 years is largely enough to equilibrate. Regarding waves, a spin up of a few days is what we typically use in all WW3 simulations.
We also want to stress that, here, we are estimating the wave impact by comparing two simulations. We thus believe that what matters the most is that all our simulations have been spun up for the same amount of time.

**P5 L12: 'Updated floe size.' how is 'floe size' defined?**
This is indeed unclear. The floe size in our study refers to the caliper diameter of the floes as defined by Rothrock and Thorndike (1984). We have added this information at the beginning of section 3 (P5L27).

**P5 L14: 'floe size is actualized'. What does this mean? Also, P9 L11: What is 'actual floe size'? Similarly P9 L25: What is the 'actual FSD'?**
We thank the reviewer for reporting these unclear expressions. The first one has been removed as the whole paragraph has been edited.
For the two other expressions mentioned, we simply removed the ambiguous word "actual", and we now refer to the FSD.

**P5 L14: 'LIM3 takes into account the WRS in its ice transport equation'. This should be stated in the Introduction, as it is a key contribution of the manuscript.**
We have added the following sentence in the last paragraph of the introduction:
P3L15 *"We focus in particular on two aspects of these interactions: firstly the effect of including the WRS, computed by the wave model, in the sea ice model, and secondly the wave-induced sea ice fragmentation and its effects on lateral melt through the addition of a FSD in the sea ice model."*

**P6 L4: How is the partial sea ice cover already accounted for in WW3?**
The estimation of sea ice-induced wave attenuation in WW3 is scaled by the sea ice concentration provided by forcing/coupling. As the WRS is directly proportional to this attenuation, it is therefore actually already scaled by the sea ice concentration. To make it clearer, we have rephrased our sentence as follows:
P6L20: "[...] *does not need to be multiplied by c, as the wave attenuation estimation in WW3 (and hence the WRS) is already scaled by the sea ice concentration to account for the partial sea ice cover."*

**P6 L22: Define the sea ice thickness distribution and the FSD function. Is the latter an areal distribution?**

In our model, the FSD is indeed an areal distribution (normalized by the cell area, just like sea ice fraction). Introduction to the sea ice thickness and floe size distribution has been added at the beginning of section 3.2. We have also added the following comment:

P8L6: *"From a technical point of view, the FSD in LIM3 is implemented as an areal distribution divided into floe size categories. It is advected in the same way as other sea ice tracers like sea ice concentration or thickness."*

**P6 L22: I think a little more explanation would be useful here for readers not familiar with the various FSD schemes in the literature. Add a sentence or so on why the Zhang et al. (2015) approach is chosen over the Horvat et Tziperman (2015) approach. The sentence from P18 L13 'assuming floes of different sizes. . .' should be stated here as well.**

In their study, Horvat et Tziperman (2015) are using a thickness and floe size joint distribution in order to represent the evolution of sea ice floes affected by a great variety of processes, not necessarily related to waves (e.g. welding, refreezing, ridging...). Zhang et al. (2015) approach is simpler and computationally cheaper, as it assumes that all floes of a given size have the same ice thickness distribution, allowing the FSD to be treated independently from the sea ice thickness distribution. To do so, they hypothesize that the FSD mostly results from the fragmentation of large unbroken floes randomly yielding floes of any smaller size than the original ones.

Here, we choose to follow the simpler approach of Zhang et al. (2015), as we only consider the effects of wave-induced sea ice fragmentation and lateral melt on the FSD evolution, and our formulation of lateral melt does not depend on sea ice thickness (Steele, 1992). We have added these comments at the beginning of section 3.2 (P7L10), along with the definitions of floe size and sea ice thickness distributions.

**P6 L28: 'implemented a FSD that enables floes to be advected..' – the FSD itself does not enable this, presumably this should say that Williams et al. (2017) implemented a scheme for advection of the FSD. Consider summarizing how this works e.g. what quantity is advected?**

This part was actually misleading and has been rephrased. In reality, Williams et al. (2017) are using a Lagrangian model, in which they advect the maximum floe size by associating it with another quantity, the "number of floes", that is assumed to be conserved. This is not directly comparable to our case and we do not think it needs to be detailed.

**P6 L31: 'We do not make any assumption on its shape in general, but the FSD is forced to follow the power-law assumed in WW3 as soon as wave-induced sea ice break-up occurs.' This sentence seems somewhat self-contradictory: there is an assumption on its shape if the FSD is constrained to follow a power-law.**

The wording is indeed awkward. The paragraph describing the implementation of the FSD has been largely re-written, following other comments from the reviewer.

**P7 L4: 'Assuming a power-law FSD is coherent with a distribution caused by a succession of break-up events (Toyota et al., 2011, Dumont et al. 2011).' The Toyota et al. 2011 study finds a change in the value of the exponent of a power-law fit to their data at around 40m. Does the model presented here assume a single power-law exponent, or include this transition?**
**Also note that the Toyota et al. 2011 study covers a small area in space and time, and therefore may not be globally applicable.**

**The Dumont et al. 2011 study itself does not show that a power-law FSD arises from a succession of break-up events, but rather provides a mathematical description for this assumption, so this citation should be removed or discussed in a different way.**

As said before, the paragraph describing the implementation of the FSD has been largely re-written. More specifically, we answer the reviewer's questions:

- Here, we assume only a single power-law exponent, as done in the studies of Dumont et al. (2011) and Williams et al. (2013). As noted by Toyota et al. (2011), the value of the exponent of the FSD found for the large floe regime that they observe is too large (>2) to be solely due to re-peated break-up of the sea ice floes. It is likely resulting from other processes, welding in partic-ular. As we do not include such processes, we do not represent this transition.

- We have added a comment to highlight that Toyota et al. (2011) study covers a small area in space and time in the discussion section:
*P18L32: "This assumption is made based on the observations analyzed by Toyota et al. (2011), that only sample a small area in time and space, so that their findings may not be applicable globally".*

- We have removed the reference to Dumont et al. (2011) here.

**P7 L8-22: The authors assume a power-law FSD in WW3 and then force LIM3 to follow the same power-law when wave fracture occurs. As they state, the effects of sea ice advection and thermodynamics cause deviations from a power-law. However, the effects of these processes may be over-ruled to continue to force the FSD to follow a power law, at a frequency determined by an arbitrary parameter. I don't understand why the authors take this approach. Why include other FSD processes if they are not always allowed to affect the FSD? How often does such over-ruling occur - is this most of the time or in a small fraction of timesteps? Is there an alternative approach to the power-law assumption? The assumption has not been well justified in the manuscript.
Similarly, can sea ice fracture be handled in the sea ice model rather than the wave model? I would have thought that this would avoid the need for the Dmax adjustment.**
Again, the part describing the implementation of the FSD and the sea ice fragmentation has been largely re-written, as we agree that the choices we have made were not justified properly. We have also added a paragraph about the choices made regarding the FSD in the discussion section. Here we also try to explain our reasoning.

- The wording of our section, with the use of the terms "forcing" and "over-ruling" was indeed a bit awkward. The right word is actually "redistribution of the FSD", just like in Zhang et al. (2015). The difference is that instead of using a redistribution scheme that will lead to power-law FSDs with a varying exponent (as the scheme used by Zhang et al. does), our scheme redistributes the FSD to make it tend towards a power law with a constant exponent. This redistribution pro-cess has indeed a strong impact on the FSD, potentially erasing the effects of advection, but so it is in nature: fragmentation by waves is an instantaneous, violent phenomenon, that completely changes the FSD (see Collins et al. (2015) for the description of a fragmentation event).

- It is difficult to quantify the number of redistributions occurring in the model as it depends on the occurrence of fragmentation events, hence on local sea ice conditions and sea states. In gen-eral, fragmentation occurrences are higher when we get closer from the sea ice edge.

- Alternative approaches for the redistribution exist, like the one suggested by Horvat et Tziperman (2015). We added a comment on this topic in the discussion (P19L1).

- Handling the sea ice fragmentation in the sea ice model is indeed an option, however it would not solve the problem raised by the reviewer of defining the value of "Dmax" from the FSD. This variable is indeed needed by the wave model to estimate the wave attenuation. The problem of how to redistribute the sea ice after fragmentation would also remain.

**P8 L3: 'This sensitivity remains really small.' This statement should be quantified more precisely, and the authors should describe how they determined this or consider adding their sensitivity results to Supplementary Material. Was sensitivity to the smallest resolved floe size tested? I would expect that lateral melt would be particularly sensitive to this.**

We have performed several simulations with different numbers of categories (from 15 to 120) and different categories widths (from 2.5 to 20m) and did not find a strong sensitivity to those parameters. The results are however much more sensitive to the choice of Dmin (see our answer below).

**P8 L16: Is this the same experiment as in Fig. 1? It would help the reader to re- state what the differences in the two runs correspond to physically. I think there are quite a few differences - evolving sea ice, advection of Dmax - which make it hard to understand what the differences in the model output mean.**

These are indeed the same experiments as those presented in Fig. 1. We have added the following sentence to make it clearer:

P10L4: *In the uncoupled WW3 simulation, Dmax evolves depending on the sea state, but sea ice thickness and concentration are constant. In the WW3-LIM3 coupled simulation, sea ice properties are all evolving as sea ice is pushed by the WRS, and Dmax is advected with the FSD in LIM3.*

**P9 L8: The Lupkes et al parametrization should be defined explicitly. Is this what LIM3 uses as standard for D in Eqn. 5? Please explain the reason for using it here.**

The Lupkes parametrization is indeed what LIM3 uses as standard for D in Eqn. 5, and we therefore aimed to compare the standard parametrization to the one we included following Horvat et Tziperman (2015), which depends on the FSD. It has been clarified in the text:

P10L29: By *default, <D>, which represents the average floe size (referred to as the caliper diameter), is a function of the sea ice concentration obtained empirically from observational data by Lupkes et al. (2012).*

**P9 L11: A situation where ice concentration is less than 0.6 and floe size is greater than 10 m could occur anywhere, for example near the ice edge in wave-free conditions, so I suggest removing the first part of this sentence.**

We actually decided to remove the whole sentence as this effect is commented in section 4.

**P9 L30: As stated above, I would expect the amount of lateral melt to depend strongly on Dmin. Have the authors investigated this? If not, the results on lateral melt should include some discussion of this.**

This is a good point. To quantify this sensitivity, we ran again the simulations described in section 3.3 for 3 different values of Dmin: 4m, 8m (the standard value), and 16m. We find that

the dependency is particularly strong when using the formula of Lupkes et al.2012. After 4 days, the quantity of sea ice volume melted laterally is more than doubled when Dmin is reduced by a factor 2 (see figure below). Using the FSD to estimate the floe size significantly reduces this sensitivity, with a value of sea ice volume melted laterally after 4 days increasing by 26% between Dmin=8m and Dmin=4m, and decreasing by 18% between Dmin=8m and Dmin=16m.

The following figure and a new paragraph have been added in Section 3.3.

**Section 4.1: This section compares the CPL and WAVE simulations at the pan-Arctic scale. The differences between the simulations include the impact of wave radiation stress and floe-size-dependent lateral melt. The authors then try to attribute various impacts to one of these two processes. Why not consider two separate runs here, one which adds the wave radiation stress only and one which adds the floe-size-dependent lateral melt only?**
**As it is, I found it difficult to understand this evaluation.**
**Section 4.1 in general was difficult to read. The text usually described differences to the CPL run. However, the WAVE and NO-CPL runs should be considered as the reference simulations, and so differences should be described in the CPL run relative to the reference runs (i.e. describe an increase in CPL relative to NO-CPL, rather than a decrease in NO-CPL). I think this would improve the readability.**
We have largely edited this section to increase the readability, and to present the NO-CPL run as the reference (as this is already done in the figures). However, we do not think that we should include additional simulations and decompose even more the inclusion of the processes in the realistic set up. Indeed, we would have to compare too many runs and the text and figures would become too long and too numerous. We do believe that the current set of simulations allows us to describe and quantify the effect of each process.

**P10 L19: It would help the reader to briefly restate the differences in the runs at the start of the paragraph, including the note at L27 ('One should keep in mind. . .). Also note mis-matched parentheses here.**
We have added a short reminder of the differences between the simulations at the beginning of this paragraph (and thus removed the note at L27).

**Sec. 4.1.3: The discussion of lateral melt would be aided by figures showing some equivalent floe size statistic from the Lupkes parametrization and from the FSD model.**
This is not straightforward due to the different natures of the floe size in these two parameterizations. When using the Lupkes parameterization, the floe size is a scalar, that cannot be directly compared to a distribution as used in the coupled simulation. Comparing the scalar with the mean floe size from the FSD would not add value here.

**P12 L11: 'This result does not reflect the fact. . .' What does this sentence mean?**
We have rephrased this sentence:
P14L6: *"This result masks the fact…"*

**P12 L17: 'Actually, in contrast to what was found in previous studies by Zhang et al. (2016), Bennetts et al. (2017), Roach et al. (2018a), de-activating completely lateral melt in both runs (not shown) has a negligible effect on the quantity of melted ice in our simulations (not shown).' The three named studies did not deactivate lateral melt, so the results presented here cannot be 'in contrast' to theirs. However, Roach, Dean, and**

**Renwick (2018) did essentially deactivate lateral melt, by setting all floe sizes to 10000m, and showed that this had no impact on sea ice concentration in the Antarctic.**
The reviewer is right that our results cannot be directly compared to these previous papers. We have removed this sentence and replaced it by a statement highlighting that compensation of lateral melt enhancement by bottom melt decrease was also reported by Roach et al. (2018) and Bateson et al. (2019) (P14L12).

**Section 4.2: This subsection is very interesting, but again hard to follow. Perhaps consider using one figure for each case, reducing the number of variables shown in figures in the main body of the paper, and moving the remainder to Supplementary Information. More figures could be added in the Supplementary for some of the 'not shown' aspects. I counted thirteen 'not shown' aspects in the paper, which seems rather high.**
We have again edited this section, trying our best to streamline the text and increase its readability. In an earlier draft of this paper we have tried to make individual figures corresponding to the different cases, but it would require more figures than we have at the moment. Moreover, we do believe that the current organization of the figures helps the reader to comprehend the differences between the different cases.
In this section specifically, most of our 'not shown' occurrences refer to the conditions before the storms… while we do believe that it should be mentioned in the text because it helps explain the difference between the cases considered, we do not think that it would add much value to the paper to show these figures in Supplementary Material.

**P16 L13: 'It is, however, mostly compensated by an increase of lateral melt.' Add that this is the converse of what has been shown in previous studies.**
The text was actually 'compensated by an increase of bottom melt', which is similar to what was found by Bateson et al. (2019), as we now mention in the text.

**P16 L8: 'The coupled model was then used to examine . . .. the effects of wave-induced sea ice break-up on sea ice melt.' Rather, the study compares their model to an alternative parametrization for lateral melt (the Lupkes parametrization), that is designed to approximate varying floe sizes for different concentrations. To isolate the impact of the wave-induced break-up, or the 'impact of the coupling' as mentioned earlier, a more suitable comparison would be to a simulation where all floes were unbroken. Otherwise, modify the discussion in the text.**
We have changed the sentence to:
 P18L6: "*(ii) the effects of using the wave-induced sea ice fragmentation to estimate lateral melt*"

**P16 L30: Similarly, the paragraph at P16 L30 compares the difference in lateral melt between the FSD model and the Lupkes parametrization (with varying floe size) to the differences found in previous studies. However, these previous studies show differences between a FSD model and a constant floe size parametrization for lateral melt, so should not be directly compared to this study. The discussion of the various studies should reflect this.**
We have modified the discussion to:
P19L22: *"Note also that we evaluate the impact of changing the lateral melt parameterization by comparing two simulations for which lateral melt depends on a varying floe size, either deduced from the FSD or estimated from the sea ice concentration using the parameterization suggested in Lüpkes et al. (2012). It differs from Zhang et al. (2016) who compare their FSD-model with a reference run without lateral melt, and from Roach et al. (2018) who use a constant floe size of*

*300 m in their lateral melt parameterization. This might partly explain the discrepancies between our respective conclusions.”*

**P17 L7: 'One should also remember that the studies of Zhang et al. (2016) and Roach et al. (2018b) were aiming at representing the evolution of floes larger than 1000 m.' This is incorrect. Both studies represent floes up to a maximum floe size of around 1000 m (radius). Also note that Roach et al. (2018a) and Roach et al. (2018b) are confused in places.**
We agree that our sentence is not accurate, and we have rephrased it as follows:
P19L18: *“One should also remember that the studies of Zhang et al. (2016) and Roach et al. (2018) aimed to represent the evolution of floes with sizes ranging from a few cm to roughly 1 km on long time scales, whereas we focus on the important processes for wave-sea ice interactions and make the assumption that unbroken floes have a uniform floe size set to 1000 m.”*

We have checked the occurrences of Roach et al. (2018a) and Roach et al. (2018b) carefully.

**P17 L13: 'Among the wave-sea ice interaction processes. . .' This sentence is unclear. Impact on what?**
We rephrased this sentence:
P19L28: *“Among the wave-sea ice interaction processes considered in this study, we find that the dynamical effect of the waves (the WRS) has a larger impact on sea ice conditions and sea surface properties than the modulation of lateral melt by sea ice fragmentation.”*

**Presentational Comments**

**Throughout, I would suggest referring to 'ocean surface waves' in the abstract and early parts of the Introduction, rather than simply 'waves' for clarity.**
**I would also suggest using 'sea ice fracture' rather than 'sea ice break-up,' as this is used in other studies**
We have replaced 'wave' by 'ocean surface waves'. Regarding the use of 'sea ice break up', we have changed it to 'sea ice fragmentation' as we do believe that it is a more realistic representation of the process occurring. This terminology was already used in previous studies (e.g. Zhang et al. 2015).

**In general, the definite article is over-used e.g. 'the sea ice near the sea ice edge' can simply be 'sea ice near the sea ice edge', 'impact the sea ice floe size' can be 'impact sea ice floe size' etc.**
We accounted for these remarks and fixed the syntax mistakes: The paper has also undergone rephrasing in many parts with the help of native speakers in order to make it clearer.

**P1 L3: 'In the present study....' - clumsy sentence, suggest rewording**
Fixed
**P1 'highlight the need to include the wave-sea ice processes in models aiming at fore-casting sea ice condition on short time scale' -> 'highlight the need to include wave-sea ice processes in models used to forecast sea ice conditions on short time scales'**
Fixed
**P2 L5: -> 'and sea ice drift'**

Fixed

**P2 L12: 'in the direction of the propagation'**

Fixed

**P2 L13: 'Southern ocean' -> 'Southern Ocean'**

Fixed

**P2 L14: 'may become more prominent in the Arctic in the future.'**

Fixed

**P2 L27: 'a first step was done' -> 'a first step was made', similarly elsewhere progress is 'made' rather than 'done'**

Fixed

**P3: reword 'wave by sea ice'; 'is implemented or not'; 'without any wind or ocean current'; also the sentences on timestep**

Fixed

**P3 L14: change 'on' to 'of'**

Fixed

**P4 L20: 'aim at compensating' -> 'was made to compensate'**

Fixed

**P4 L31 'in this particular year'**

Fixed

**P4 L31: reword 'storms occurring during it'**

Fixed

**P5 L1: 'referred to as WAVE'**

Fixed

**P5 L9: sentences about average thickness - seem to use a lot of words to say some-thing fairly straightforward**

Fixed

**P5 L29: define vector k**

Fixed

**P7 L8: 'the coupling between the two models can be done' -> 'the two models can be coupled'**

Fixed

**P10 L4: The introduction to Section 4 seems unnecessarily lengthy and should be made more concise.**

Fixed

**P10 L5: 'the impact of the including the wave-sea ice interactions' - reword**

Fixed: The sentence has been changed to "*in order to quantify the impact of the coupling on wave, sea ice and ocean surface properties*"

**P13 L25: 'that is exposed upwind (and waves)' - reword**

Fixed (removing "and waves")

**P14 L19: 'could in principle modified' - reword**

The whole sentence has actually been edited.

**P14 L34 'very high waves of which attenuation induces WRS' - reword**

The sentence has been changed to *"the strong storm generates high waves, inducing a WRS as large as the wind stress close to the sea ice where most of the attenuation takes place."*

**P15 L14: 'pattern than' -> 'pattern to'**

Fixed

**P15 L32 'low concentrated' -> 'of low concentration'**

Fixed

**P16 L15: 'generating higher and more energetic waves'**

Fixed

**P17 L7: 'were aiming at representing' -> 'aimed to represent'**

Fixed

**P17 L14: 'additional lateral source melt' - reword**

Fixed

**Section 4.1 figures – in the reference plots, I found the colormaps rather counter-intuitive. Consider choosing maps that are white at zero.**

Fixed

**Fig. 7: y-axis label lists the units as %, but values on the y-axis are out of 1. I presume that the 10ˆ2 kmˆ3 corresponds the numbers on the figure, but this should be noted in the legend.**

Fixed

**#Referee 3:**

**Anonymous Referee #3**
**The manuscript Toward a coupled model to investigate wave-sea ice interactions in the Arctic marginal ice zone by Boutin et al. presents a model that couples waves and sea ice dynamics to study the impact of waves on sea ice evolution over the Arctic Ocean. The model includes a floe size and thickness distribution as a prognostic variable that is exchanged between the sea ice and wave components. The FSTD obeys an evolution equation that includes floe-size dependent processes such as lateral melt and wave break-up. A focus is put on the wave radiation stress arising from wave attenuation in sea ice that imposes an additional force on the ice, and on the floe-size dependent lateral melt parameterization. The impact of wave-related processes on sea ice are studied by comparing simulations of NEMO-LIM3 (ice-ocean component) that is coupled and uncoupled to WW3 (wave component) over a pan-Arctic domain, and during two storm case. The comparison is done over a month-long period, at the end of summer 2010, after a 8-year spin-up period. Overall the paper makes a significant contribution to the modeling of polar marine environment in the sense that it provides a very useful tool to study the complexities wave-ice interactions and their impact over different spatio-temporal scales. The discussion puts the study in the context of the recent developments and describes the limitations, thus pointing towards important issues to be addressed in order to make further progress (duration of the simulation, atmospheric and oceanic coupling, floe-size dependent ice rheology missing, freezing period not studied, etc.). It is well written, despite some typos and corrections that need to be made, and descriptions of model implementation and results are detailed enough, although some key information is missing (see below). It is thus worthy of publication, after minor revisions are made.**

We thank the reviewer for their careful reading of our manuscript and for their comments and suggestions. We have tried to address their questions and concerns, as detailed in the following. In our comments, PXLY refers to page X line Y of the attached updated manuscript.

**Specific comments**

**P4. L18. Wave attenuation is a central piece of the study, as it determines the wave radiation stress and, to a certain extent, the extent of the wave-induced ice break-up area (i.e. the marginal ice zone). Because of this, I suggest that in addition to referring to Ardhuin et al. (2018) for the choice of the wave attenuation, authors recall the main characteristics of the attenuation scheme. Is it floe-size and/or thickness dependent, and how? Is it a dissipative or scattering scheme (or a mix of both)? This could be done in a few lines.**
We have added a short description of the processes described in Ardhuin et al. (2018) and Boutin et al. (2018) in P3L27.

**P6. L1. Another central piece of the study is the ice drift resulting from the momentum balance. Here the WRS is added as an external forcing term that will be balanced by the internal stress, and model solutions may depend strongly on rheology**

**parameters. I understand that this term (rheology) has not been modified significantly from what's typically used by LIM3 users, and that studying the ie rheology is not the focus of the paper, but it needs to be described minimally here. The rheology contains a few parameters that can be tuned for various reasons, including the compressive strength, the shear-to-compressive strength ratio, if not the yield curve itself or the numerical scheme. Describe what rheology is used and what are the main parameter values. Maybe adding a table would serve well that purpose.**

We have added a few comments along with details on the parameters used in the rheology in section 2:
P3L32*: "The model includes a standard Elasto-Visco-Plastic rheology (Hunke and Dukowicz, 1997), using the stress tensor formulation of Bouillon et al. (2013) adapted for the C-grid used in the model. The ice strength is determined following Hibler III (1979), with the ice strength P following $P = P^* h e^{C(1-c)}$, where $P^*$=20,000 N/m2 and C=20 are empirical positive parameters, and h is the cell-average sea ice thickness. The plastic failure threshold lies on an elliptical yield curve of which eccentricity is set equal to 2. The number of sub-time steps used to solve the momentum equation is set to 120."*

**P11. L14. Warmer and saltier surface waters in the CPL run seems to point towards that enhanced turbulent mixing arising by increased shear stress between the ice and the ocean, dominates over enhanced melting, which tends to produce fresh and cold anomalies. The following section focuses on an interpretation of that response in terms of the differences between the lateral melt parameterization. Have you looked at mixing as a possible mechanism for explaining it? Are there anomalies in the mixing or mixed layer depth in the marginal ice zone? This mechanism is discussed very clearly later in the two storm cases, but it would be interesting to discuss it also for the pan-Arctic case.**
We had a look at the differences in mixed layer depth and properties, but the signal was very patchy, making it difficult to draw conclusions at the pan-Arctic scale. Indeed, we do believe that local conditions matter a lot (e.g., the relative directions of the wind, sea ice, waves, and surface currents) in determining the impact of the waves, which motivated us to investigate regional cases.

**P19. Eq A3. Define $D_*$. And later, define also $n_*$. Is $D_*$ equivalent to $D_{n*}$?**

Explicit definitions of these terms have been added in the appendix (P21L18,P22L6).

**Some typos**

**P5. L14. Replace actualized by updated.**
Fixed
**P5. L20....is transferred to what has caused this attenuation.**
Fixed
**P5. Eq2. Remove parentheses around σ.**
Fixed

**P6. L22. multi-category.**
Fixed
**P7. L29.c has already been introduced as the concentration earlier.**
We think a reminder might help the reader there.
**P8. L10. Is Toyota et al. (2011) the right reference for this statement? There are older and more appropriate references for this it seems. The smallest floe size that can be generated by flexural break-up is thickness-dependent. Maybe this should be acknowledged.**
We now cite Mellor (1986) instead. This study suggests a formulation for this lower limit of floe size that can break due to flexural break-up.The fact that this lower limit is thickness dependent is true but adding it to the text might add confusion in our opinion, as this paragraph focuses on the definition of floe size categories with constant upper and lower limits.

**P8. L25. Uncoupled instead of not coupled (also at various other place in the manuscript).**
Fixed
**P9. L8. Based on a number of observations.**
Fixed
**P9. L17. Rather than on sea ice conditions.**
Fixed (with concentration instead of conditions)
**P10. L8....on sea ice conditions.**
Fixed
**P10. L19. There is no panel e on Fig. 5.**
Fixed
**P11. L9. Do you refer to the grid cell average thickness?**
Yes, we edited so that it is now clearly specified.
**P11. L11. There are also differences…**
Fixed
**P12. L1....property anomalies.**
Fixed (we kept the word *difference* to keep coherency with the rest of the text)
**P14. L6. Difference (singular).**
Fixed
**P17. L24. when trying to forecast…**
Fixed
**Fig2. Schematic summary of...The two boxes correspond…**
Fixed
**Fig3. Panel c. notcpl should be replaced by NOT_CPL in the index. You can also specify the run elsewhere than in the index to avoid expanding indices.**
Fixed
**Fig5. The black and grey contours…**
Fixed

[revised manuscript text omitted]

---

## Referee Report (RR1)

The authors' revisions have improved the quality of the manuscript. However, some elements of the model they developed still need to be clarified before publication. The comments marked with * are the most important to address.

Questions regarding the model

P5L26: 'LIM3 provides sea ice floe size, thickness and concentration to WW3.' Is this floe size a grid-cell average? If so, how is the average computed?

P5L30: 'WW3 then returns the WRS to LIM3, as well as the updated floe size if fragmentation has occurred.' Is this floe size passed from WW3 to LIM3 the maximum floe size Dmax?

P5L33 and P8L14: 'If fragmentation has occurred in the wave model, the FSD in LIM3 is re-arranged, transferring sea ice from large floe categories to smaller floe categories. The resulting FSD obeys a power-law similar to the one assumed in WW3.' And 'our redistribution is set to a power-law with a constant exponent as soon as fragmentation occurs.' My interpretation of the wave fracture scheme is as follows: 'We assume wave fracture results in a truncated power law defined for floe sizes up to some calculated Dmax, under the constraint of conservation of area. Dmax represents the largest size of floes remaining after the fracture event.' Is this correct? If so, I think it should be described in this way, rather than using worlds like 'set' or 'force' or 'rearrange.'

Eq. 6: The integral of the FSD from zero to infinity is 1. Is the integral from Dmin to infinity the ice concentration, c? If so, is the FSD modified by any process that affects c? (For example, loss of ice area by basal melt).

P8L20: What is the value of the constant exponent?

P9L5: 'Noisy Dmax distributions' and 'smoother FSDs' - are these in space, or across floe size, or in time? Why are smoother FSDs desirable? It is argued earlier that wave fracture is a violent event.

*P10L6: 'Dmax is advected with the FSD in LIM3.' However, Dmax is not an area-conserved quantity, so should not be advected as an area-conserved tracer - see Horvat & Tziperman (2017). This should be discussed in the text. It is still not clear to me why Dmax needs to be defined in LIM3 if LIM3 contains a FSD. If wave fracture results in a truncated power law defined by a parameter Dmax in WW3, then the FSD in LIM3 will be zero in categories with a size greater than Dmax. The FSD can be advected in LIM3 and there is no need to advect Dmax.

Eq. 12: I presume that a similar equation, without the integral over floe sizes, is used to evolve the FSD under lateral melt - this should be clarified in the text.

*P11L13: 'Note also that Horvat and Tziperman (2015) and Roach et al. (2018) are considering floe radii in their study, while we are working with floe diameters (hence adding a factor of 2 in Eq.12).' Horvat & Tziperman (2015) also have a factor two in their equation for lateral melt using floe radius. Should you have a second factor two?

Typographical comments:

P2L22: `Most of the recent efforts' - replace with 'several recent efforts'

P2L32: 'floe size distribution' should not be capitalised

P3L10: 'forcing a wave model by sea ice properties' -> 'forcing a wave model with sea ice properties'

P5L21: 'we remove the first 3 days' -> 'we exclude the first 3 days from the analysis'?

P5L31 'LIM3 takes into account the WRS in its ice transport equation, and advects the sea ice and its information on floe size. This information is carried by a newly implemented FSD, the sea ice concentration being distributed among floe size categories.' I don't think that 'this information is carried by a FSD' makes sense. LIM3 advects the FSD, which is defined as an areal distribution.

P7L1 'It is therefore necessary to exchange information on floe size between the two models, which can be done by using a FSD.' Similarly, the exchange of information between the two models is not 'done' by the FSD, rather the FSD is exchanged between the two models (or parameters defining the FSD are exchanged - this should be clarified).

P9L20: 'really small' -> 'very small'

P14L12: References to Tsamados et al. (2015) and Roach, Dean & Renwick (2018) should be added when discussing basal/lateral melt compensation.

P21L21: 'In WW3, a fragmentation event occurs if, firstly, waves with a wavelength λ apply a strain on sea ice greater than a given threshold, and secondly if λ/2 which is assumed to be the value of the new maximum floe size is lower than the current D max value in the wave model' -> Do you mean that both conditions are required for wave fracture, or only one?

---

## Author Response (AR2)

**The authors' revisions have improved the quality of the manuscript. However, some elements of the model they developed still need to be clarified before publication. The comments marked with * are the most important to address.**

We thank the reviewer for their careful reading of our manuscript and for their comments and suggestions. We have tried to address their questions and concerns in our response.

**Questions regarding the model**

The next 3 comments concern the same paragraph that was indeed unclear.

**P5L26: 'LIM3 provides sea ice floe size, thickness and concentration to WW3.' Is this floe size a grid-cell average? If so, how is the average computed?**

This floe size is the maximum floe size.

**P5L30: 'WW3 then returns the WRS to LIM3, as well as the updated floe size if fragmentation has occurred.' Is this floe size passed from WW3 to LIM3 the maximum floe size Dmax?**
It is indeed the maximum floe size.

**P5L33 and P8L14: 'If fragmentation has occurred in the wave model, the FSD in LIM3 is rearranged, transferring sea ice from large floe categories to smaller floe categories. The resulting FSD obeys a power-law similar to the one assumed in WW3.' And 'our redistribution is set to a power-law with a constant exponent as soon as fragmentation occurs.' My interpretation of the wave fracture scheme is as follows: 'We assume wave fracture results in a truncated power law defined for floe sizes up to some calculated Dmax, under the constraint of conservation of area. Dmax represents the largest size of floes remaining after the fracture event.' Is this correct? If so, I think it should be described in this way, rather than using worlds like 'set' or 'force' or 'rearrange.'**

This interpretation is correct. We have rephrased following the reviewer's suggestion.
In response to these 3 comments, we have rewritten the paragraph as follows:

P5L24: 'The objective of this section is to present the theoretical background and the practical implementation of the coupling between LIM3 and WW3. Fig. 2 shows the principle of the coupling and the variables that are exchanged between the two models. Briefly, LIM3 sends the sea ice thickness, concentration, and maximum floe size (estimated from the newly implemented FSD) to WW3. This maximum floe size, referred to as Dmax, represents the largest size of floes remaining after the fragmentation event. These quantities are used by WW3 in order to estimate the wave attenuation and wave-induced sea ice fragmentation. Note that, in

general, we refer to sea ice thickness as the cell-average of the sea ice thickness distribution, $g_h$, which is used as a state variable in LIM3. In the coupling, we actually exchange the ice-cover average sea ice thickness, although this choice does not significantly affect our results. WW3 then returns the WRS to LIM3, as well as the updated maximum floe size if fragmentation has occurred. The occurrence of fragmentation is thus determined in the wave model, depending on the wave conditions (see Boutin et al., 2018). In LIM3, we assume wave-induced fragmentation results in a truncated power law defined for floe sizes up to the maximum floe size estimated in WW3. LIM3 takes into account the WRS in its ice transport equation, and advects the sea ice and its FSD, which is defined as an areal distribution. The FSD in LIM3 is also used to estimate lateral melt.'

We also rephrased the sentence P8L14:

P8L28: '[...], we assume that wave-induced fragmentation results in a truncated power law with a constant exponent defined for floe sizes up to the value of Dmax received from WW3, under the constraint of conservation of area.'

**Eq. 6: The integral of the FSD from zero to infinity is 1. Is the integral from Dmin to infinity the ice concentration, c? If so, is the FSD modified by any process that affects c? (For example, loss of ice area by basal melt).**

The integral from Dmin to infinity the ice concentration is indeed equal to *c.* The shape of the FSD, however, is only modified by lateral melt and sea ice fragmentation, its shape being unaffected by all the other processes driving the evolution of *c*. We have added a sentence to clarify this:
P8L13: 'In our implementation, the only processes affecting the FSD are lateral melt and sea ice fragmentation. Other processes driving the evolution of the sea ice concentration do not modify the shape of the FSD.'

**P8L20: What is the value of the constant exponent?**
The exponent corresponds to a value of about -1.85 if we consider the cumulative floe size distribution, which is the distribution generally used to represent the FSD in the scientific literature (e.g. Toyota et al., 2011, Herman et al., 2018). This value originates from the parameterization of wave-induced fragmentation by Dumont et al. (2011), and is the one used in WW3 by Boutin et al. (2018). We have added this comment in the text (P8L30).

**P9L5: 'Noisy Dmax distributions' and 'smoother FSDs' - are these in space, or across floe size, or in time? Why are smoother FSDs desirable? It is argued earlier that wave fracture is a violent event.**

Noisy and smoother are indeed not the appropriate terminology that should be used here. Dmax represents a floe size associated with the largest floes still present after a fragmentation event.

When setting $k_{Dmax}=1$, a negligible reduction in the proportion of large floes in the FSD after a fragmentation event (when lateral melt occurs for instance) results in an immediate reduction of Dmax. This can be problematic, as reducing the value of Dmax during a fragmentation event allows for the wave to propagate further, generating more fragmentation. Values of $k_{Dmax}$ between 0.5 and 0.8 allow for little variation in the FSD while keeping the same value of Dmax, thus giving more weight to fragmentation than to other processes in the estimation of Dmax. We rewrote our comment to clarify our motivations concerning the tuning of $k_{Dmax}$ (P9L20).

**\*P10L6: 'Dmax is advected with the FSD in LIM3.' However, Dmax is not an area-conserved quantity, so should not be advected as an area-conserved tracer - see Horvat & Tziperman (2017). This should be discussed in the text. It is still not clear to me why Dmax needs to be defined in LIM3 if LIM3 contains a FSD. If wave fracture results in a truncated power law defined by a parameter Dmax in WW3, then the FSD in LIM3 will be zero in categories with a size greater than Dmax. The FSD can be advected in LIM3 and there is no need to advect Dmax.**

This formulation is indeed unclear. In LIM3, what is advected is the FSD. There is no advection of the quantity Dmax. However, the variable used to exchange information between LIM3 and WW3 is Dmax. This is because in WW3, the information on the level of fragmentation of sea ice is contained in Dmax. To send information on the sea ice fragmentation to WW3, LIM3 must therefore send an estimate of Dmax to WW3. To do so, we suggested a way to compute a value of Dmax from the FSD. The value of Dmax in LIM3 may therefore change due to the advection of the FSD. We have therefore rephrased the text as follows:

P10L23: '[...] the FSD is advected in LIM3.'

A discussion on why Dmax cannot be advected and why it must be estimated in LIM3 has been added at the beginning of section 3.2.

P7L8: 'As Dmin is assumed to be constant in WW3, the FSD in the wave model is thus only a function of Dmax. Ideally, WW3 would therefore send the value of Dmax to the sea ice model, where it would be advected and updated due to the effects of thermodynamical and mechanical processes, and then sent back to WW3. Yet, Dmax is not an area-conserved quantity, and therefore cannot be advected as a tracer (Williams et al., 2017). Instead, we thus choose to define a FSD in LIM3, from which a maximum floe size can be estimated and then sent to the wave model.'

**Eq. 12: I presume that a similar equation, without the integral over floe sizes, is used to evolve the FSD under lateral melt - this should be clarified in the text.**

We are not sure to fully understand this comment. The term responsible for the redistribution of the FSD due to lateral melt is Phi_th, and it is explicitly described here. In Eq. 12, we show the whole integral to illustrate the difference in the lateral melt estimation with the "standard" parameterization. To make this clearer, we have rewritten the description of the terms in Eq. 12 as follows:

P11L27: 'where $\phi_{th}$ is the change in the FSD due to lateral melt (see Eq.7)'

**\*P11L13: 'Note also that Horvat and Tziperman (2015) and Roach et al. (2018) are considering floe radii in their study, while we are working with floe diameters (hence adding a factor of 2 in Eq.12).' Horvat & Tziperman (2015) also have a factor two in their equation for lateral melt using floe radius. Should you have a second factor two?**

We should indeed have a second factor two. This factor two comes from the fact that Horvat and Tziperman (2015) and Roach et al. (2018) represent the evolution of the floe radius, and therefore include in their equation a term they call $G_r$ representing dr/dt, the rate at which the floe radius changes due to lateral growth/melt. In our case, we represent the evolution of the floe diameter, and the rate at which it changes due to lateral growth/melt is equal to 2dr/dt. We added these clarifications to our comment P11L28.

**Typographical comments:**
**P2L22: `Most of the recent efforts' - replace with 'several recent efforts'**
We have rephrased this.
**P2L32: 'floe size distribution' should not be capitalised**
We have rephrased this.
**P3L10: 'forcing a wave model by sea ice properties' -> 'forcing a wave model with sea ice properties'**
We have rephrased this.
**P5L21: 'we remove the first 3 days' -> 'we exclude the first 3 days from the analysis'?**
We have rephrased this.
**P5L31 'LIM3 takes into account the WRS in its ice transport equation, and advects the sea ice and its information on floe size. This information is carried by a newly implemented FSD, the sea ice concentration being distributed among floe size categories.' I don't think that 'this information is carried by a FSD' makes sense. LIM3 advects the FSD, which is defined as an areal distribution.**
The whole paragraph has been rephrased following this comment and previous ones. The formulation is now:
P6L1: "LIM3 takes into account the WRS in its ice transport equation, and advects the sea ice and its FSD, which is defined as an areal distribution."

**P7L1 'It is therefore necessary to exchange information on floe size between the two models, which can be done by using a FSD.' Similarly, the exchange of information between the two models is not 'done' by the FSD, rather the FSD is exchanged between the two models (or parameters defining the FSD are exchanged - this should be clarified).**
We have rephrased this

**P9L20: 'really small' -> 'very small'**
We have rephrased this

**P14L12: References to Tsamados et al. (2015) and Roach, Dean & Renwick (2018) should be added when discussing basal/lateral melt compensation.**
We agree and added these references as suggested by the reviewer.

**P21L21: 'In WW3, a fragmentation event occurs if, firstly, waves with a wavelength λ apply a strain on sea ice greater than a given threshold, and secondly if λ/2 which is assumed to be the value of the new maximum floe size is lower than the current D max value in the wave model' -**
**> Do you mean that both conditions are required for wave fracture, or only one?**
Both conditions are required for wave fracture. We rephrased this sentence as:

[revised manuscript text omitted]

---

## Author Response (AR3)

We thank the editor for his careful reading of our answers to referee #2 comments.
You will find below our answer to the editor comment:

**Would you please confirm that the correction to floe diameter in Eq 12 is purely typographical, and has no implication for the calculations that you present? (Response to point below from Referee 2.)**

**P11L13: 'Note also that Horvat and Tziperman (2015) and Roach et al. (2018) are considering floe radii in their study, while we are working with floe diameters (hence adding a factor of 2 in Eq.12).' Horvat & Tziperman (2015) also have a factor two in their equation for lateral melt using floe radius. Should you have a second factor two?**

We confirm that the correction to floe diameter in Eq 12 is purely typographical. This factor 2 was missing in the first version of our manuscript but has always been taken into account in our model.
The factor 2 in Horvat & Tziperman (2015) arises from the cylindrical derivative term in their computation of the rate of change of the cumulative number of floes with respect to time, while the factor 2 we added in the equation comes from the time rate of change of the diameter (which is twice the one of the radius). Our formulation and the one of Horvat & Tziperman (2015) are equivalent, the only difference being the floe size variable we consider.

We do not include a marked-up manuscript version in this response since no change has been done to the manuscript.